# Embrace the Gap: VAEs Perform Independent Mechanism Analysis

**Patrik Reizinger**[\*,1], **Luigi Gresele**[\*,2], **Jack Brady**[\*,1], **Julius von Kügelgen**[2,3], **Dominik Zietlow**[2,4], **Bernhard Schölkopf**[2], **Georg Martius**[2], **Wieland Brendel**[1], **and Michel Besserve**[†,2]

[1]University of Tübingen, Germany
[2]Max Planck Institute for Intelligent Systems, Tübingen, Germany
[3]University of Cambridge, Cambridge, United Kingdom
[4]Amazon Web Services, Tübingen, Germany
{patrik.reizinger,jack.brady,wieland.brendel}@uni-tuebingen.de,
{luigi.gresele,jvk,bs,gmartius,besserve}@tue.mpg.de, zietld@amazon.de

## Abstract

Variational autoencoders (VAEs) are a popular framework for modeling complex data distributions; they can be efficiently trained via variational inference by maximizing the evidence lower bound (ELBO), at the expense of a gap to the exact (log-)marginal likelihood. While VAEs are commonly used for disentangled representation learning, it is unclear why ELBO maximization would yield such representations, since unregularized maximum likelihood estimation generally cannot invert the data-generating process without additional assumptions. Yet, VAEs often succeed at this task. We seek to elucidate this apparent paradox by studying nonlinear VAEs in the limit of near-deterministic decoders. We first prove that, in this regime, the optimal encoder approximately inverts the decoder—a commonly used but unproven conjecture—which we refer to as *self-consistency*. Leveraging self-consistency, we show that the ELBO converges to a regularized log-likelihood. This allows VAEs to perform what has recently been termed independent mechanism analysis (IMA): it adds an inductive bias towards decoders with column-orthogonal Jacobians, which helps recovering the true latent factors. The gap between ELBO and log-likelihood is therefore welcome, since it bears unanticipated benefits for nonlinear representation learning. In experiments on synthetic and image data, we show that VAEs uncover the true latent factors when the data generating process satisfies the IMA assumption.

## 1 Introduction

Latent Variable Models (LVMs) allow to effectively approximate a complex data distribution and to sample from it [3, 48]. Deep LVMs employ a neural network (the *decoder* or *generator*) to parameterize the conditional distribution of the observations given latent variables, which are typically assumed to be independent. However, Maximum Likelihood Estimation (MLE) of the model parameters is computationally intractable. In *Variational Autoencoders (VAEs)* [35, 56], the exact log-likelihood is substituted with a tractable lower bound, the evidence lower bound (ELBO). This objective introduces an approximate posterior of the latents given the observations (the *encoder*) from a suitable variational distribution whose mean and covariance are parametrized by neural networks. The encoder is introduced to efficiently train a deep LVM: however, it is not explicitly designed to extract useful representations [17, 58].

---

[\*]Equal contribution. Code available at: github.com/rpatrik96/ima-vae
[†]Senior author

36th Conference on Neural Information Processing Systems (NeurIPS 2022).

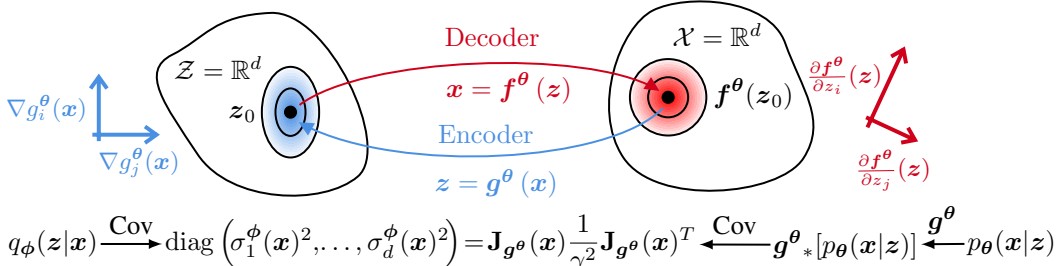

$$q_{\phi}(\boldsymbol{z}|\boldsymbol{x}) \xrightarrow{\text{Cov}} \text{diag}\left(\sigma_1^{\phi}(\boldsymbol{x})^2, \ldots, \sigma_d^{\phi}(\boldsymbol{x})^2\right) = \mathbf{J}_{\boldsymbol{g}^{\theta}}(\boldsymbol{x})\frac{1}{\gamma^2}\mathbf{J}_{\boldsymbol{g}^{\theta}}(\boldsymbol{x})^T \xleftarrow{\text{Cov}} \boldsymbol{g}^{\theta}{}_{*}[p_{\boldsymbol{\theta}}(\boldsymbol{x}|\boldsymbol{z})] \xleftarrow{\boldsymbol{g}^{\theta}} p_{\boldsymbol{\theta}}(\boldsymbol{x}|\boldsymbol{z})$$

Figure 1: **Modeling choices in VAEs promote *Independent Mechanism Analysis (*IMA*) [23].** We assume a Gaussian VAE (3), and prove that in the near-deterministic regime the mean encoder approximatetely inverts the mean decoder, $\boldsymbol{g}^{\theta} \approx \boldsymbol{f}^{\theta-1}$ (*self-consistency*, Prop. 1). **Bottom:** Closing the gap requires matching the covariances of the variational (LHS, $q_{\phi}(\boldsymbol{z}|\boldsymbol{x})$) and the true posterior (RHS, approximated by $\boldsymbol{g}^{\theta}{}_{*}[p_{\boldsymbol{\theta}}(\boldsymbol{x}|\boldsymbol{z})]$, cf. § 3.2 for details). Under self-consistency, an encoder with diagonal covariance enforces a row-orthogonal encoder Jacobian $\mathbf{J}_{\boldsymbol{g}^{\theta}}(\boldsymbol{x})$—or equivalently, a column-orthogonal decoder Jacobian $\mathbf{J}_{\boldsymbol{f}^{\theta}}(\boldsymbol{z})$. This regularization was termed Independent Mechanism Analysis (IMA) [23] and shown to be beneficial for learning the true latent factors. The connection elucidates unintended benefits of using the ELBO for representation learning.

Nonetheless, VAEs and their variants are widely used in representation learning [25, 1], where they often recover semantically meaningful representations [39, 8, 34, 5]. Our understanding of this empirical success is still incomplete, since (deep) LVMs with independent latents are nonidentifiable from i.i.d. data [29, 42]; different models fitting the data equally well may yield arbitrarily different representations, thus making the recovery of a ground truth generative model impossible. While auxiliary variables, weak supervision [28, 31, 21, 43, 72, 19], or specific model constraints [29, 67, 68, 26, 23] can help identifiability, the mechanism through which the ELBO may enforce a useful inductive bias remains unclear, despite recent efforts [5, 57, 38, 15, 71].

In this work, we investigate the benefits of optimizing the ELBO for representation learning by analyzing VAEs in a *near-deterministic* limit for the conditional distribution parametrized by the nonlinear decoder. Our first result concerns the encoder's optimality in this regime. Previous works relied on the intuitive assumption that the encoder inverts the decoder in the optimum [50, 38, 71]; we formalize this *self-consistency* assumption and prove its validity for the optimal variational posterior in the near-deterministic nonlinear regime.

Using self-consistency, we show that the ELBO tends to a regularized log-likelihood—rather than to the exact one as conjectured in previous work [50]. The regularization term allows VAEs to perform what has been termed Independent Mechanism Analysis (IMA) [23]: it encourages column orthogonality of the decoder's Jacobian. This generalizes previous findings based on linearizations or approximations of the ELBO [57, 44, 38], and allows us to characterize the gap w.r.t. the log-likelihood in the deterministic limit. Our results elucidate the gap between ELBO and exact log-likelihood as a possible mechanism through which the ELBO implements a useful inductive bias. Unlike the unregularized log-likelihood, the IMA-regularized objective can help invert the data generating process under suitable assumptions [23]. We verify this by training VAEs in experiments on synthetic and image data, showing that they can recover the ground truth factors when the IMA assumptions are met.

The **contributions** of this paper can be summarized as follows:
- we characterize and prove *self-consistency* of VAEs in the near-deterministic regime (i.e., when the decoder variance tends to zero), justifying its usage in previous works (§ 3.1);
- we show that under self-consistency, the ELBO converges to a regularized log-likelihood (§ 3.2), and discuss its possible role as a useful inductive bias in representation learning;
- we test the applicability of our theoretical results in experiments on synthetic and image data, and show that VAEs recover the true latent factors when the IMA assumptions are met (§ 4).

## 2 Background

We will connect two unsupervised learning objectives: the ELBO in VAEs and the IMA-regularized log-likelihood. Both stem from LVMs with latent variables $\boldsymbol{z}$ distributed according to a *prior* $p_0(\boldsymbol{z})$, and a mapping from $\boldsymbol{z}$ to observations $\boldsymbol{x}$ given by a conditional generative model $p_{\boldsymbol{\theta}}(\boldsymbol{x}|\boldsymbol{z})$.

**Variational Autoencoders.** Optimizing the data likelihood $p_{\boldsymbol{\theta}}(\boldsymbol{x})$ in deep LVMs—i.e., finding decoder parameters $\boldsymbol{\theta}$ maximizing $\int p_{\boldsymbol{\theta}}(\boldsymbol{x}|\boldsymbol{z})p_0(\boldsymbol{z})d\boldsymbol{z}$—is intractable in general, so approximate objectives are required. Variational approximations [63] replace the true posterior $p_{\boldsymbol{\theta}}(\boldsymbol{z}|\boldsymbol{x})$ by an approximate one, called the *variational posterior* $q_{\boldsymbol{\phi}}(\boldsymbol{z}|\boldsymbol{x})$, which is a stochastic mapping $\boldsymbol{x} \mapsto \boldsymbol{z}$ with parameters $\boldsymbol{\phi}$. This allows to evaluate a tractable evidence lower bound (ELBO) [35, 56] of the model's log-likelihood that can be defined as

$$\text{ELBO}(\boldsymbol{x}, \boldsymbol{\theta}, \boldsymbol{\phi}) = \mathbb{E}_{q_{\boldsymbol{\phi}}(\boldsymbol{z}|\boldsymbol{x})}\left[\log p_{\boldsymbol{\theta}}(\boldsymbol{x}|\boldsymbol{z})\right] - \text{KL}\left[q_{\boldsymbol{\phi}}(\boldsymbol{z}|\boldsymbol{x})||p_0(\boldsymbol{z})\right]. \tag{1}$$

The two terms in (1) are sometimes interpreted as a reconstruction term measuring the sample quality of the decoder and a regularizer—the Kullback-Leibler Divergence (KL) between the prior and the encoder [36]. The variational approximation trades off computational efficiency with a difference w.r.t. the exact log-likelihood, which is expressed alternatively as (see [17, 36] and Appx. A)

$$\text{ELBO}(\boldsymbol{x}, \boldsymbol{\theta}, \boldsymbol{\phi}) = \log p_{\boldsymbol{\theta}}(\boldsymbol{x}) - \text{KL}\left[q_{\boldsymbol{\phi}}(\boldsymbol{z}|\boldsymbol{x})||p_{\boldsymbol{\theta}}(\boldsymbol{z}|\boldsymbol{x})\right], \tag{2}$$

where the KL between variational and true posteriors characterizes the *gap*: if the variational family of $q_{\boldsymbol{\phi}}(\boldsymbol{z}|\boldsymbol{x})$ does not include $p_{\boldsymbol{\theta}}(\boldsymbol{z}|\boldsymbol{x})$, the ELBO will be strictly smaller than $\log p_{\boldsymbol{\theta}}(\boldsymbol{x})$.

VAEs [35] rely on the variational approximation in (1) to train deep LVMs where neural networks parametrize the *encoder* $q_{\boldsymbol{\phi}}(\boldsymbol{z}|\boldsymbol{x})$ and the *decoder* $p_{\boldsymbol{\theta}}(\boldsymbol{x}|\boldsymbol{z})$. A common modeling choice constrains the variational family of $q_{\boldsymbol{\phi}}(\boldsymbol{z}|\boldsymbol{x})$ to a factorized Gaussian with posterior means $\mu_k^{\boldsymbol{\phi}}(\boldsymbol{x})$ and variances $\sigma_k^{\boldsymbol{\phi}}(\boldsymbol{x})^2$ for the $k^{th}$ factor $z_k|\boldsymbol{x}$, and with a diagonal covariance $\boldsymbol{\Sigma}_{\boldsymbol{z}|\boldsymbol{x}}^{\boldsymbol{\phi}}$; and the decoder to a factorized Gaussian, conditional on $\boldsymbol{z}$, with mean $\boldsymbol{f}^{\boldsymbol{\theta}}(\boldsymbol{z})$ and an isotropic covariance in $d$ dimensions,

$$z_k|\boldsymbol{x} \sim \mathcal{N}(\mu_k^{\boldsymbol{\phi}}(\boldsymbol{x}), \sigma_k^{\boldsymbol{\phi}}(\boldsymbol{x})^2); \qquad \boldsymbol{x}|\boldsymbol{z} \sim \mathcal{N}\left(\boldsymbol{f}^{\boldsymbol{\theta}}(\boldsymbol{z}), \gamma^{-2}\mathbf{I}_d\right). \tag{3}$$

**The deterministic limit of VAEs.** The stochasticity of VAEs makes it nontrivial to relate them to generative models with deterministic decoders such as Independent Component Analysis (see paragraph below), though postulating a deterministic regime (where the decoder precision $\gamma^2$ becomes infinite) is possible. Interestingly, Nielsen et al. [50] explored this deterministic limit and argued that *deterministic* VAEs optimize an exact log-likelihood, similar to normalizing flows [55, 51]. Normalizing flows model arbitrarily complex distributions using a simple base distribution $p_0(\boldsymbol{z})$ and nonlinear, *deterministic and invertible* transformations $\boldsymbol{f}^{\boldsymbol{\theta}}$. Through a change of variables,[3] the likelihood of the original variables becomes

$$\log p_{\boldsymbol{\theta}}(\boldsymbol{x}) = \log p_0(\boldsymbol{z}) - \log\left|\mathbf{J}_{\boldsymbol{f}^{\boldsymbol{\theta}}}(\boldsymbol{z})\right|. \tag{4}$$

The comparison is nontrivial, since VAEs contain an encoder and a decoder, whereas normalizing flows consist of a single architecture. Nielsen et al. [50] made this analogy by resorting to what we call a *self-consistency assumption*, stating that the VAE encoder inverts the decoder. We define self-consistency in the *near-deterministic* regime: as the decoder variance goes to zero, i.e. $\gamma \to +\infty$.

**Definition 1** ((Near-deterministic) self-consistency). *For a fixed $\boldsymbol{\theta}$, assume that mean decoder $\boldsymbol{f}^{\boldsymbol{\theta}}$ is invertible with inverse $\boldsymbol{g}^{\boldsymbol{\theta}}$, and that a map associates each choice of decoder parameters and observation $(\boldsymbol{\theta}, \gamma, \boldsymbol{x})$ to an encoder parameter $(\boldsymbol{\theta}, \gamma, \boldsymbol{x}) \mapsto \widehat{\boldsymbol{\phi}}(\boldsymbol{\theta}, \gamma, \boldsymbol{x})$, we say the VAE is self-consistent whenever*

$$\boldsymbol{\mu}^{\widehat{\boldsymbol{\phi}}}(\boldsymbol{x}) \to \boldsymbol{g}^{\boldsymbol{\theta}}(\boldsymbol{x}) \quad \text{and} \quad \boldsymbol{\sigma}^{\widehat{\boldsymbol{\phi}}}(\boldsymbol{x})^2 \to \mathbf{0} \text{ , as } \gamma \to +\infty. \tag{5}$$

The encoder parameter map $\widehat{\boldsymbol{\phi}}$ reflects the choice of a particular encoder model for each $(\boldsymbol{\theta}, \gamma)$ pair:[4] in § 3.1, we study this problem by introducing and justifying a particular choice for $\widehat{\boldsymbol{\phi}}$ (see also § 5). This self-consistency assumption appears central to deterministic claims [50, 38], but has not yet been proven. In particular, Nielsen et al. [50] assume that taking the deterministic limit is well-behaved. However, VAEs' *near-deterministic* properties have not been investigated analytically.

**Identifiability, ICA, and IMA.** Independent Component Analysis (ICA) [9, 30] models observations as the *mixing* of a latent vector $\boldsymbol{z}$ with independent components through a deterministic function $\boldsymbol{f}$, i.e., $\boldsymbol{x} = \boldsymbol{f}(\boldsymbol{z}), p_0(\boldsymbol{z}) = \prod_i p_0(z_i)$.[5] In ICA the focus is on defining conditions under which the original

---

[3] note that in normalizing flows the change of variables is usually expressed in terms of $\boldsymbol{g}^{\boldsymbol{\theta}} = \boldsymbol{f}^{\boldsymbol{\theta}-1}$

[4] both the ELBO and $\widehat{\boldsymbol{\phi}}$ depends on the decoder precision $\gamma$: we will omit this in the following for simplicity

[5] the conditional distribution $p(\boldsymbol{x}|\boldsymbol{z})$ is therefore degenerate

latent variables can be recovered from observations—i.e., the model is "identifiable by design" [31]. The goal is to learn an unmixing $\boldsymbol{g^\theta}$ such that the recovered components $\boldsymbol{y} = \boldsymbol{g^\theta}(\boldsymbol{x})$ are estimates of the true ones up to some ambiguities (e.g., permutation and element-wise nonlinear transformations). Unfortunately, the nonlinear problem is nonidentifiable without further constraints [16, 29]: any two observationally equivalent models can yield components which are arbitrarily entangled, thus making recovery of the ground truth factors impossible. This is typically shown by suitably constructed counterexamples [29, 42], and it was argued to imply impossibility statements for unsupervised disentanglement [42, 65]. Identifiability can be recovered when *auxiliary* variables [31, 21, 33, 19] are available, or exploiting a temporal structure in the data [28, 24].

Restrictions on the mixing function class (e.g., linear [9]) are another possibility to recover identifiability [29, 67]. Recently, Gresele et al. [23] proposed restricting the function class by taking inspiration from the *principle of independent causal mechanisms* [52], in an approach termed Independent Mechanism Analysis (IMA). IMA postulates that the latent components influence the observations "independently", where influences correspond to the partial derivatives $\partial \boldsymbol{f^\theta} / \partial z_k$, and their non-statistical independence amounts to an orthogonality condition. While full identifiability has not been proved for this model class, it was shown to rule out classical families of spurious solutions used as counterexamples to identifiability of unconstrained non-linear ICA [23, 4]. Moroever, Buchholz et al. [4] further demonstrated local identifiability of this function class. Also, IMA constraints were empirically shown [23, 62] to help recover the ground truth through regularization of the log-likelihood in (4) with an objective $\mathcal{L}_{\text{IMA}}(\boldsymbol{f^\theta}, \boldsymbol{z}) := \log p_{\boldsymbol{\theta}}(\boldsymbol{x}) - \lambda \cdot c_{\text{IMA}}(\boldsymbol{f^\theta}, \boldsymbol{z})$, where $\lambda > 0$ and the regularization term $c_{\text{IMA}}(\boldsymbol{f^\theta}, \boldsymbol{z})$ and its expectation $C_{\text{IMA}}(\boldsymbol{f^\theta}, p_0)$ are given by

$$c_{\text{IMA}}(\boldsymbol{f^\theta}, \boldsymbol{z}) = \sum_{k=1}^{d} \log \left\| \frac{\partial \boldsymbol{f^\theta}}{\partial z_k}(\boldsymbol{z}) \right\| - \log \left| \mathbf{J}_{\boldsymbol{f^\theta}}(\boldsymbol{z}) \right|; \quad C_{\text{IMA}}(\boldsymbol{f^\theta}, p_0) = \mathbb{E}_{p_0(\boldsymbol{z})} \left[ c_{\text{IMA}}(\boldsymbol{f^\theta}, \boldsymbol{z}) \right], \quad (6)$$

and termed *local* (resp. *global*) IMA contrast. When $\boldsymbol{f^\theta}$ is in the IMA function class (i.e., $C_{\text{IMA}}(\boldsymbol{f^\theta}, p_0)$ vanishes), the objective is equal to the log-likelihood; otherwise, it lower bounds it.

## 3 Theory

Our theoretical analysis assumes that all the model's defining densities $(p_0(\boldsymbol{z}), q_{\boldsymbol{\phi}}(\boldsymbol{z}|\boldsymbol{x})$ and $p_{\boldsymbol{\theta}}(\boldsymbol{x}|\boldsymbol{z}))$ are factorized. We also assume a Gaussian decoder, matching common modeling practice in VAEs.

**Assumption 1** (Factorized VAE class with isotropic Gaussian decoder and log-concave prior)**.** *We are given a fixed latent prior and three parameterized classes of $\mathbb{R}^d \to \mathbb{R}^d$ mappings: the mean decoder class $\boldsymbol{\theta} \mapsto \boldsymbol{f^\theta}$, and the mean and standard deviation encoder classes, $\boldsymbol{\phi} \mapsto \boldsymbol{\mu^\phi}$ and $\boldsymbol{\phi} \mapsto \boldsymbol{\sigma^\phi}$ s.t.*

- *(i) $p_0(\boldsymbol{z}) \sim \prod_k m(z_k)$, with $m$ being smooth and fully supported on $\mathbb{R}$, having bounded non-positive second-order, and bounded third-order logarithmic derivatives;*
- *(ii) the encoder and decoder are of the form in (3), with isotropic decoder covariance $1/\gamma^2 \mathbf{I}_d$;*
- *(iii) the variational mean and variance encoder classes are universal approximators;*
- *(iv) for all $\boldsymbol{\theta}$, $\boldsymbol{f^\theta} : \mathbb{R}^d \to \mathbb{R}^d$ is a bijection with inverse $\boldsymbol{g^\theta}$, and both are $C^2$ with bounded first and second order derivatives.*

Crucially, *both the mean encoder and the mean decoder can be nonlinear*. Moreover, the family of log-concave priors contains the commonly-used Gaussian distribution as a special case. We study the *near-deterministic decoder* regime of such models, where $\gamma \to +\infty$. This regime is expected to model data generating processes with vanishing observation noise well—in line with the typical ICA setting—and is commonly considered in theoretical analyses of VAEs, e.g., in [50] (which additionally assumes quasi-deterministic encoders), and in [44, 38]. Unlike Nielsen et al. [50], we consider a large but finite $\gamma$, not *at* the limit $\gamma = \infty$, where the decoder is fully deterministic. In fact, for any large but finite $\gamma$, the objective is well-behaved and amenable to theoretical analysis, while the KL-divergence is undefined in the deterministic setting. The requirement in assumption *(iv)* deviates from common practice in VAEs—where observations are typically higher-dimensional—but it allows to connect VAEs and exact likelihood methods such as normalizing flows [50] (see also § 5).

Due to considering $\gamma \to +\infty$, results are stated in the following "big-O" notation for an integer $p$:

$$f(\boldsymbol{x}, \gamma) = g(\boldsymbol{x}, \gamma) + O_{\gamma \to +\infty}(1/\gamma^p) \iff \gamma^p \| f(\boldsymbol{x}, \gamma) - g(\boldsymbol{x}, \gamma) \| \text{ is bounded as } \gamma \to +\infty \,.$$

### 3.1 Self-consistency

In this section, we will prove a *self-consistency* result in the near-deterministic regime. This rests on characterizing optimal variational posteriors (i.e., those minimizing the ELBO gap w.r.t. the likelihood) for a *particular point $x$* and *fixed decoder parameters $\boldsymbol{\theta}$*. Based on (2), any associated optimal choice of encoder parameters satisfies

$$\widehat{\boldsymbol{\phi}}(\boldsymbol{x}, \boldsymbol{\theta}) \in \arg\max_{\boldsymbol{\phi}} \mathrm{ELBO}(\boldsymbol{x}; \boldsymbol{\theta}, \boldsymbol{\phi}) = \arg\min_{\boldsymbol{\phi}} \mathrm{KL}\left[q_{\boldsymbol{\phi}}(\boldsymbol{z}|\boldsymbol{x})||p_{\boldsymbol{\theta}}(\boldsymbol{z}|\boldsymbol{x})\right] . \tag{7}$$

We call *self-consistent* ELBO the resulting achieved value, denoted as

$$\mathrm{ELBO}^*(\boldsymbol{x}; \boldsymbol{\theta}) = \mathrm{ELBO}(\boldsymbol{x}; \boldsymbol{\theta}, \widehat{\boldsymbol{\phi}}(\boldsymbol{x}, \boldsymbol{\theta})) . \tag{8}$$

The expression in (7) corresponds to a problem of *information projection* [10, 48] of $p_{\boldsymbol{\theta}}(\boldsymbol{z}|\boldsymbol{x})$ onto the set of factorized Gaussian distributions. This means that given a variational family, we search for the optimal $q_{\boldsymbol{\phi}}(\boldsymbol{z}|\boldsymbol{x})$ to minimize the KL to $p_{\boldsymbol{\theta}}(\boldsymbol{z}|\boldsymbol{x})$. While such information projection problems are well studied for closed convex sets where they yield a unique minimizer [11], the set projected onto in our case is not convex (convex combinations of arbitrary Gaussians are not Gaussian), making this problem of independent interest. After establishing upper and lower bounds on the KL divergence (exposed in Prop. 7-8 in Appx. C.2), we obtain the following self-consistency result.

**Proposition 1.** *[Self-consistency of near-deterministic VAEs] Under Assumption 1, for all $\boldsymbol{x}, \boldsymbol{\theta}$, as $\gamma \to +\infty$, there exists at least one global minimum solution of (7). These solutions satisfy*

$$\boldsymbol{\mu}^{\widehat{\boldsymbol{\phi}}}(\boldsymbol{x}) = \boldsymbol{g}^{\boldsymbol{\theta}}(\boldsymbol{x}) + O(1/\gamma) \quad and \quad \sigma_k^{\widehat{\boldsymbol{\phi}}}(\boldsymbol{x})^2 = O(1/\gamma^2) , \text{ for all } k . \tag{9}$$

Prop. 1 states that minimizing the ELBO gap (equivalently, maximizing the ELBO) w.r.t. the encoder parameters $\boldsymbol{\phi}$ implies in the limit of large $\gamma$ that the encoder's mean $\boldsymbol{\mu}^{\widehat{\boldsymbol{\phi}}}(\boldsymbol{x})$ tends to $\boldsymbol{g}^{\boldsymbol{\theta}}(\boldsymbol{x})$, the image of $\boldsymbol{x}$ by the *inverse* decoder. We can interpret this as the decoder "inverting" the encoder. Additionally, the variances of the encoder will converge to zero.

Let us now consider the relevance of this result for training VAEs, i.e., maximizing the expectation of the ELBO for an observed distribution $p(\boldsymbol{x})$. While maximization *only* w.r.t. $\boldsymbol{\phi}$ in (7) does not match common practice—which is learning $\boldsymbol{\theta}$ and $\boldsymbol{\phi}$ *jointly*—it models this process in the limit of large-capacity encoders. Indeed, in this case, (7) can be solved for each $\boldsymbol{x}$ as a separate learning problem, which entails that the following inequality is satisfied for any parameter choice

$$\mathbb{E}_{\boldsymbol{x} \sim p(\boldsymbol{x})}\left[\mathrm{ELBO}(\boldsymbol{x}; \boldsymbol{\theta}, \boldsymbol{\phi})\right] = \int p(\boldsymbol{x})\mathrm{ELBO}(\boldsymbol{x}; \boldsymbol{\theta}, \boldsymbol{\phi})d\boldsymbol{x}$$
$$\leq \int p(\boldsymbol{x})\mathrm{ELBO}(\boldsymbol{x}; \boldsymbol{\theta}, \widehat{\boldsymbol{\phi}}(\boldsymbol{x}, \boldsymbol{\theta}))d\boldsymbol{x} =: \mathbb{E}_{\boldsymbol{x} \sim p(\boldsymbol{x})}\left[\mathrm{ELBO}^*(\boldsymbol{x}; \boldsymbol{\theta})\right] . \tag{10}$$

The joint optimization of encoder and decoder parameters thus reduces to optimizing the subset of pairs $(\boldsymbol{\theta}, \widehat{\boldsymbol{\phi}}(\boldsymbol{x}, \boldsymbol{\theta}))$, and is equivalent to optimizing the expected self-consistent ELBO, that is

$$\underset{\boldsymbol{\theta}, \boldsymbol{\phi}}{\mathrm{maximize}} \, \mathbb{E}_{\boldsymbol{x} \sim p(\boldsymbol{x})}\left[\mathrm{ELBO}(\boldsymbol{x}; \boldsymbol{\theta}, \boldsymbol{\phi})\right] \iff \underset{\boldsymbol{\theta}}{\mathrm{maximize}} \, \mathbb{E}_{\boldsymbol{x} \sim p(\boldsymbol{x})}\left[\mathrm{ELBO}^*(\boldsymbol{x}; \boldsymbol{\theta})\right] \tag{11}$$

This problem reduction is aligned with the original purpose of the ELBO: building a tractable but optimal likelihood approximation. Namely, (i) ELBO$^*$ depends on the same parameters as the likelihood ($\boldsymbol{x}, \gamma$ and $\boldsymbol{\theta}$), (ii) its gap KL $\left[q_{\boldsymbol{\phi}}(\boldsymbol{z}|\boldsymbol{x})||p_{\boldsymbol{\theta}}(\boldsymbol{z}|\boldsymbol{x})\right]$ is minimal. The problem reduction of (11) allows us to compare the optimality of different decoders and Prop. 1 helps addressing the case of near-deterministic decoders.

### 3.2 Self-consistent ELBO, IMA-regularized log-likelihood and identifiability of VAEs

We want to investigate how the choice of $q_{\boldsymbol{\phi}}(\boldsymbol{z}|\boldsymbol{x})$ and $p_{\boldsymbol{\theta}}(\boldsymbol{x}|\boldsymbol{z})$ implicitly regularizes the Jacobians of their means $\boldsymbol{\mu}^{\widehat{\boldsymbol{\phi}}}(\boldsymbol{x})$ and $\boldsymbol{f}^{\boldsymbol{\theta}}(\boldsymbol{z})$ in the near-deterministic regime. Exploiting self-consistency, we are able to precisely characterize how this happens: we formalize this in Thm. 1.

**Theorem 1.** *[VAEs with a near-deterministic decoder approximate the IMA objective] Under Assumption 1, the variational posterior satisfies*

$$\sigma_k^{\widehat{\boldsymbol{\phi}}}(\boldsymbol{x})^2 = \left(-\frac{d^2 \log p_0}{dz_k^2}(g_k^{\boldsymbol{\theta}}(\boldsymbol{x})) + \gamma^2 \left\|\left[\mathbf{J}_{\boldsymbol{f}^{\boldsymbol{\theta}}}\left(\boldsymbol{g}^{\boldsymbol{\theta}}(\boldsymbol{x})\right)\right]_{:k}\right\|^2\right)^{-1} + O(1/\gamma^3) , \tag{12}$$

*and the self-consistent ELBO (10) approximates the IMA-regularized log-likelihood (6):*

$$\mathrm{ELBO}^*(\boldsymbol{x}; \boldsymbol{\theta}) = \log p_{\boldsymbol{\theta}}(\boldsymbol{x}) - c_{\mathrm{IMA}}(\boldsymbol{f}^{\boldsymbol{\theta}}, \boldsymbol{g}^{\boldsymbol{\theta}}(\boldsymbol{x})) + O_{\gamma \to \infty}(1/\gamma^2) . \tag{13}$$

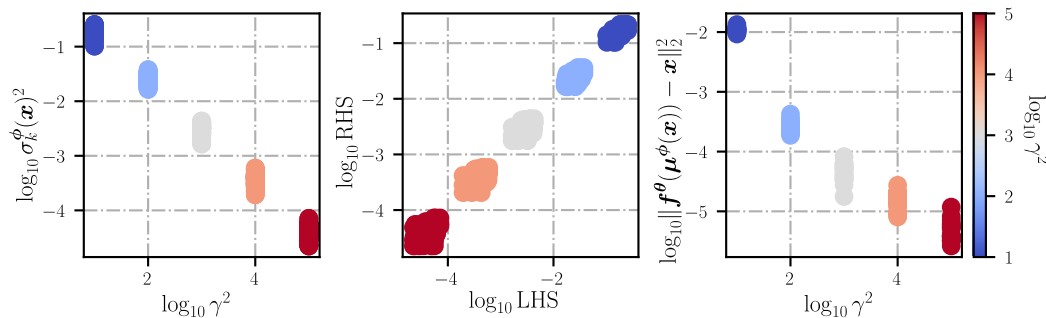

Figure 2: Self-consistency (Prop. 1) in VAE training, on a log-log plot, cf. 4.1 for details. **Left**: convergence of $\sigma_k^{\widehat{\phi}}(\boldsymbol{x})^2$ to 0; **Center:** connecting $\sigma_k^{\widehat{\phi}}(\boldsymbol{x})^2$, $\gamma^2$, and the column norms of the decoder Jacobian via LHS and RHS of (12); **Right:** convergence of $\boldsymbol{\mu}^{\widehat{\phi}}(\boldsymbol{x})$ to $\boldsymbol{g}^{\boldsymbol{\theta}}(\boldsymbol{x})$

Proof is in Appx. B. Below, we provide a qualitative argument on the interplay between distributional assumptions in the VAE and implicit constraints on the decoder's Jacobian and its inverse.

**Modeling assumptions implicitly regularize the mean decoder class $\boldsymbol{f}^{\boldsymbol{\theta}}$ under self-consistency.** In the near deterministic regime, $p_{\boldsymbol{\theta}}(\boldsymbol{x})$ gets close to the pushforward distribution of the prior by the mean decoder $\boldsymbol{f}^{\boldsymbol{\theta}}{}_{*}[p_0(\boldsymbol{z})]$, which can be used to show that the true posterior $p_{\boldsymbol{\theta}}(\boldsymbol{z}|\boldsymbol{x}) = p_{\boldsymbol{\theta}}(\boldsymbol{x}|\boldsymbol{z})p_0(\boldsymbol{z})/p_{\boldsymbol{\theta}}(\boldsymbol{x})$ is approximately the pushforward through the inverse mean decoder $\boldsymbol{g}^{\boldsymbol{\theta}}{}_{*}[p_{\boldsymbol{\theta}}(\boldsymbol{x}|\boldsymbol{z})]$ (see Appx. A for more details). If we select a given latent $\boldsymbol{z}_0$ and denote its image by $\boldsymbol{f}^{\boldsymbol{\theta}}(\boldsymbol{z}_0)$, then we can locally linearize $\boldsymbol{g}^{\boldsymbol{\theta}}$ by its Jacobian $\mathbf{J}_{\boldsymbol{g}^{\boldsymbol{\theta}}} = \mathbf{J}_{\boldsymbol{g}^{\boldsymbol{\theta}}}(\boldsymbol{f}^{\boldsymbol{\theta}}(\boldsymbol{z}_0))$, yielding a Gaussian for the pushforward distribution $\boldsymbol{g}^{\boldsymbol{\theta}}{}_{*}[p_{\boldsymbol{\theta}}(\boldsymbol{x}|\boldsymbol{z})]$ with covariance $1/\gamma^2\mathbf{J}_{\boldsymbol{g}^{\boldsymbol{\theta}}}\mathbf{J}_{\boldsymbol{g}^{\boldsymbol{\theta}}}^T$. As the sufficient statistics of a Gaussian are given by its mean and covariance, the structure of the posterior covariance $\boldsymbol{\Sigma}_{\boldsymbol{z}|\boldsymbol{x}}^{\boldsymbol{\phi}}$ (which is by design diagonal, cf. (3)) is crucial for minimizing the gap in (2). Practically, this implies that in the zero gap limit, the covariances of $q_{\boldsymbol{\phi}}(\boldsymbol{z}|\boldsymbol{x})$ and $p_{\boldsymbol{\theta}}(\boldsymbol{z}|\boldsymbol{x})$ should match, i.e., $1/\gamma^2\mathbf{J}_{\boldsymbol{g}^{\boldsymbol{\theta}}}\mathbf{J}_{\boldsymbol{g}^{\boldsymbol{\theta}}}^T$ will be diagonal with entries $\sigma_k^{\boldsymbol{\phi}}(\boldsymbol{x})^2$ and therefore $\mathbf{J}_{\boldsymbol{g}^{\boldsymbol{\theta}}}$ has orthogonal rows. We can express the decoder Jacobian via the inverse function theorem as $\mathbf{J}_{\boldsymbol{f}^{\boldsymbol{\theta}}}(\boldsymbol{z}_0) = \mathbf{J}_{\boldsymbol{g}^{\boldsymbol{\theta}}}(\boldsymbol{f}^{\boldsymbol{\theta}}(\boldsymbol{z}_0))^{-1}$. As the inverse of a row-orthogonal matrix has orthogonal columns, $\boldsymbol{f}^{\boldsymbol{\theta}}$ satisfies the IMA principle. Additionally, we can relate the variational posterior's variances to the column-norms of $\mathbf{J}_{\boldsymbol{f}^{\boldsymbol{\theta}}}$ as $\sigma_k^{\boldsymbol{\phi}}(\boldsymbol{x})^2 = 1/\gamma^2\|\left[\mathbf{J}_{\boldsymbol{f}^{\boldsymbol{\theta}}}(\boldsymbol{z}_0)\right]_{:k}\|^{-2}$, as predicted by (12).

Our argument indicates that minimizing the gap between the ELBO and the log-likelihood encourages column-orthogonality in $\mathbf{J}_{\boldsymbol{f}^{\boldsymbol{\theta}}}$ by matching the covariances of $q_{\boldsymbol{\phi}}(\boldsymbol{z}|\boldsymbol{x})$ and $\boldsymbol{g}^{\boldsymbol{\theta}}{}_{*}[p_{\boldsymbol{\theta}}(\boldsymbol{x}|\boldsymbol{z})]$. When $q_{\boldsymbol{\phi}}(\boldsymbol{z}|\boldsymbol{x}) = p_{\boldsymbol{\theta}}(\boldsymbol{z}|\boldsymbol{x})$, the gap is closed; this is only possible if the decoder is in the IMA class, for which $c_{\text{IMA}}$ vanishes and the ELBO *tends to an exact log-likelihood*. To the best of our knowledge, we are the first to prove this for nonlinear functions, extending related work for linear VAEs [44].

**Implications for identifiability of VAEs.** While previous works argued that the VAE objective favors decoders with a column-orthogonal Jacobian [57, 38], they did not exactly characterize how: our result shows that the self-consistent ELBO tends to a regularized log-likelihood, where the regularization term $c_{\text{IMA}}$ explicitly enforces this (soft) constraint. Thus, it possibly explains why VAEs are successful in learning disentangled representations: namely, the IMA function class provably rules out certain spurious solutions for nonlinear ICA [23], and the IMA-regularized log-likelihood was empirically shown to be beneficial in recovering the true latent factors. Thus, we speak about *embracing the gap*, as its functional form equips VAEs with a useful inductive bias. While the IMA function class has not yet been shown to be identifiable in the classical sense such results exist for special cases such as conformal maps ($d = 2$ [29], generalized by the very recent work in [4]), isometries [26] and for closely-related unsupervised nonlinear ICA models [69]. Moreover, Buchholz et al. [4] demonstrate a *local* form of identifiability for the IMA function class. In the following, we empirically corroborate that VAEs: 1) recover the ground truth sources when the mixing satisfies IMA, and thereby 2) achieve unsupervised disentanglement.

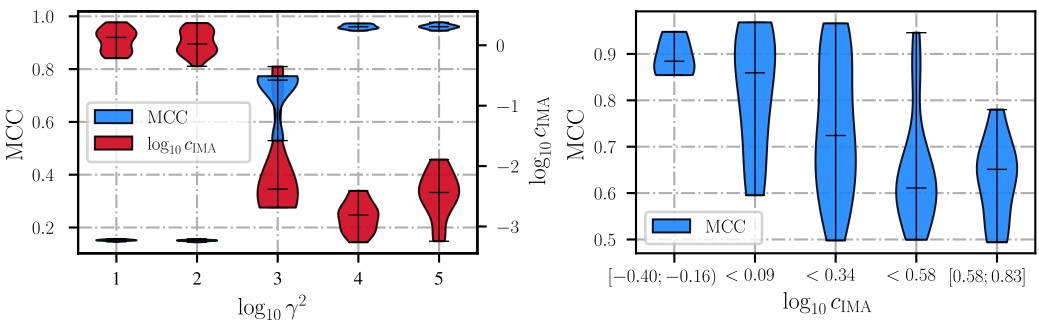

Figure 3: **Left:** $c_{\text{IMA}}$ and Mean Correlation Coefficient (MCC) for 3-dimensional Möbius mixings **Right:** MCC depending on the *volume-preserving linear map's* $c_{\text{IMA}}$ ($\gamma^2 = 1\text{e}5$)

## 4   Experiments

Our experiments serve three purposes: 1) demonstrating that self-consistency holds in practice (§ 4.1); 2) showing the relationship of the self-consistent ELBO*, the IMA-regularized and unregularized log-likelihood objectives (§ 4.2); and 3) providing empirical evidence that the connection to the IMA function class in VAEs can lead to success in learning disentangled representations (§ 4.3). More details are provided in Appx. F.

### 4.1   Self-consistency in practical conditions

**Experimental setup.** We use a 3-layer Multi-Layer Perceptron (MLP) with smooth Leaky ReLU nonlinearities [22] and orthogonal weight matrices—which intentionally does not belong to the IMA class, as our results are more general. The 60,000 source samples are drawn from a standard normal distribution and fed into a VAE composed of a 3-layer MLP encoder and decoder with a Gaussian prior. We use 20 seeds for each $\gamma^2 \in \{1\text{e}1; 1\text{e}2; 1\text{e}3; 1\text{e}4; 1\text{e}5\}$.
**Results.** Fig. 2 summarizes our results, featuring the *logarithms* on each axes. The **left** plot shows that the posterior variances $\sigma_k^\phi(\boldsymbol{x})^2$ converge to zero with a $1/\gamma^2$ rate, as predicted by (9). The **center** plot shows that the expression for $\sigma_k^\phi(\boldsymbol{x})^2$ corresponds to (12) in the optimum of the ELBO by comparing both sides of the equation. The **right** plot shows approximate convergence of the mean encodings $\boldsymbol{\mu}^{\widehat{\phi}}(\boldsymbol{x})$ to $\boldsymbol{g}^\theta(\boldsymbol{x})$ with a $1/\gamma$ rate (see § 5). As $\boldsymbol{f}^\theta$ is not guaranteed to be invertible, we use instead the *optimal* encoder and decoder parameters to compare $\boldsymbol{f}^\theta(\boldsymbol{\mu}^{\widehat{\phi}}(\boldsymbol{x}))$ to $\boldsymbol{x}$.

### 4.2   Relationship between ELBO*, IMA-regularized, and unregularized log-likelihoods

**Experimental setup.** We use an MLP $\boldsymbol{f}^\theta$ with square upper-triangular weight matrices and invertible element-wise nonlinearities to construct a mixing not in the IMA class [23] and fix the VAE decoder to the ground truth such that (4) gives the true data log-likelihood. This way, we ensure that the unregularized and IMA-regularized log-likelihoods differ and make the claim of Nielsen et al. [50] comparable to ours. With a fixed decoder, the ELBO* depends only on $\phi$, therefore we only train the encoder with $\gamma^2$ values from [1e1; 1e5] (5 seeds each).
**Results.** Fig. 4 compares the difference of the estimate of ELBO* and the unregularized/IMA-regularized log-likelihoods after convergence over the whole dataset. As the decoder and the data are fixed, $\log p_\theta(\boldsymbol{x})$ and $C_{\text{IMA}}$ will not change during training, only the ELBO* does. The figure shows that as $\gamma \to +\infty$, ELBO* approaches $\mathcal{L}_{\text{IMA}}(\boldsymbol{f}^\theta, \boldsymbol{z})$, as predicted by Thm. 1, and not $\log p_\theta(\boldsymbol{x})$, as stated in [50]—the difference is $C_{\text{IMA}}$.

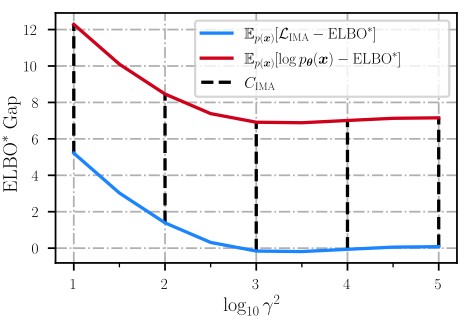

Figure 4: Comparison of the ELBO*, the IMA-regularized and unregularized log-likelihoods over different $\gamma^2$. Error bars are omitted as they are orders of magnitudes smaller

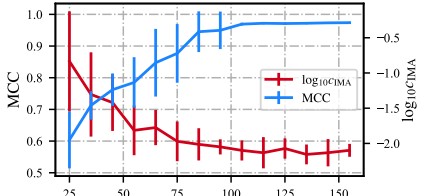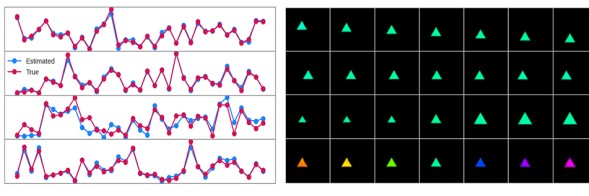

Figure 5: **Left:** $c_{\text{IMA}}$ and MCC for Sprites [66] during training ($\gamma^2 = 1$); **Center:** true and estimated latent factors for the best trained VAE on Sprites; **Right:** the corresponding latent interpolations and MCC values (from top to bottom): $y$- (0.989), $x$-position (0.996), scale (0.933), and color (0.989)

### 4.3 Connecting the IMA principle, $\gamma^2$, and disentanglement

**Experimental setup (synthetic).** We use 3-dimensional conformal mixings (i.e., the Möbius transform [53]) from the IMA class with *uniform* ground-truth and prior distributions. Our results quantify the relationship of the decoder Jacobian's IMA-contrast and identifiability with MCC [27] and show how this translates to disentanglement—we note that MCC was already used to quantify disentanglement [72, 37]. To determine whether a mixing from the IMA class is beneficial for disentanglement, we apply a volume-preserving linear map after the Möbius transform (using 100 seeds) to make $c_{\text{IMA}} \neq 0$. Other parameters are the same as in § 4.1, with the exception of picking the best $\gamma^2 = 1\text{e}5$.

**Results (synthetic).** The **left** of Fig. 3 empirically demonstrates the benefits of optimizing the IMA-regularized log-likelihood. By increasing $\gamma^2$, MCC increases, while $c_{\text{IMA}}$ decreases, suggesting that VAEs in the near-deterministic regime encourage disentanglement by enforcing the IMA principle. The **right** plot shows that when the mixing is outside the IMA class, MCC decreases, corroborating the benefits of IMA class mixings for disentanglement.

**Experimental setup (image).** We train a VAE (not $\beta$-VAE) with a factorized Gaussian posterior and Beta prior on a Sprites image dataset generated using the spriteworld renderer [66] with a Beta ground truth distribution. Similar to [32], we use four latent factors, namely, *x- and y-position, color and size*, and omit factors that can be problematic, such as shape (as it is discrete) and rotation (due to symmetries) [57, 37]. Our choice is motivated by [26, 18] showing that this data-generating process may approximately satisfy the IMA principle.

**Results (image).** The **left** of Fig. 5 indicates that VAEs can learn the true latent factors and MCC is *anticorrelated* with $c_{\text{IMA}}$, reinforcing the hypothesis that the data-generating process belongs to the IMA class. The **center** plot compares estimated and true latent factors from the best model (scaling and permutation indeterminacies are removed), whereas the **right** plot shows the corresponding latent interpolations—thus, connecting identifiability (measured by MCC) to disentanglement.

## 5 Limitations

**The near-deterministic regime.** Our theory relies on $\gamma \to +\infty$; this is the regime where posterior collapse may be avoided [44], and where calculating the reconstruction loss may be possible even without sampling [38]. However, in practice it may be unclear when $\gamma^2$ is large enough. This seems to be problem-dependent [57, 44], and possibly tied to the covariance of the observations [60, 59]. Moreover, large values of $\gamma^2$ may be harder to optimize due to an exploding reconstruction term in (1). This may be one explanation for the slight deviation of Fig. 2, right from our theory's predictions: while convergence of $\boldsymbol{\mu}^\phi(\boldsymbol{x})$ to $\boldsymbol{g}^\theta$ matches the prediction in Prop. 1, its rate is not precisely the one predicted for the self-consistent ELBO (10). Another cause could be the encoder's finite capacity. Nonetheless, we have experimentally shown that for realistic hyperparameters, VAEs' behavior matches the predictions of our theory for the near-deterministic regime.

**Dimensionality.** The setting in § 3 requires equal dimensionality for observations $\boldsymbol{x}$ and latents $\boldsymbol{z}$, in line with work on normalizing flows [51] and nonlinear ICA [28, 31, 24] (but see, e.g., [33]). For high-dimensional images, however, it is often assumed that $\boldsymbol{x}$ lives on a lower-dimensional manifold embedded in a higher-dimensional space, where the dimensionality of $\boldsymbol{x}$ is greater than $\boldsymbol{z}$ [13]. While our theoretical results do not cover this case, we observe empirically in Fig. 5 that the predictions of our theory remain accurate when observations are high-dimensional images. Extending our theory to this setting could leverage ideas explored in, e.g., [13, 12, 7] and is left for future work.

**The ELBO, the self-consistent ELBO, and amortized inference.** There are in principle multiple ways to obtain self-consistency (Defn. 1). Notably, one could simply force the variational mean and variance encoder maps to behave this way; unlike [38], we model the actual behavior of VAEs trained under ELBO maximization, and obtain self-consistency as a result. For this, we assume that the optimal encoder, which minimizes the gap between ELBO and log-likelihood, can be learned. This is not guaranteed in general, since it requires universal approximation capability of the encoder. On the other hand, (10) requires *unamortized* inference to introduce ELBO*, which does not depend on $\phi$. As in practice amortized inference may be used to efficiently estimate a single set of $\phi$ for all $x$ [61], it can lead to a suboptimal gap to the log-likelihood and discrepancies with our theoretical predictions.

## 6 Discussion

**On disentanglement in unsupervised VAEs.** It is widely believed that unsupervised VAEs cannot learn disentangled representations [42, 33], motivating work on models with, e.g., conditional priors [33] or sparse decoding [47]. We show that under certain assumptions, ELBO optimization can implement useful inductive biases for representation learning, yielding disentangled representations in unsupervised VAEs. However, while our results are formulated for VAEs, some of the most successful models at disentanglement are modifications thereof—e.g., $\beta$-VAEs [25, 5], with an additional parameter $\beta$ multiplying the KL in (1). While they deviate from the information projection setting considered in § 3.1, their objectives are equivalent to the ELBO in a sense described in Appx. A.3, which allows us to derive convergence to the IMA-regularized likelihood objective for $\gamma/\sqrt{\beta} \to +\infty$. This encompasses the deterministic limit, and also the setting $\beta \to 0$ with constant $\gamma$ described in [38]. Whether this theoretical regime matches common practice remains an open question. Overall, we stress that we uncover *one* possible mechanism through which VAEs may achieve disentanglement. By connecting to IMA [23], we discuss implications on recovering the ground truth under suitable assumptions, extending uniqueness results presented in [38]. We speculate that our success in disentanglement is probably due to selecting data sets where the mixing is in the IMA class (cf. [26, 18]), which presumably was not the case in [42].

**Characterizing the ELBO gap for nonlinear models.** Thm. 1 characterizes the gap between ELBO and true log-likelihood for nonlinear VAEs, and extends the linear analysis of Lucas et al. [44] and the results of Dai et al. [14] in the affine case; we also empirically characterize the gap in the deterministic limit in § 4.2. An unanticipated consequence of this result is that—consistent with [44]—VAEs optimize the IMA-regularized log-likelihood in the near-deterministic limit, and not the unregularized one, as stated in [50].

**Extensions to related work.** Several papers discuss the (near-)deterministic regime [50, 57, 38, 13]. For example, Nielsen et al. [50] postulate a deterministic VAE with the encoder inverting the decoder. Also Kumar and Poole [38] work in that regime, but without justifying the relationship between the encoder and decoder. Although they show that the choice of $p_0(z)$ and $q_\phi(z|x)$ influences uniqueness (by, e.g., ruling out rotations), this does not imply recovering the true latents. Our approach formalizes (Defn. 1), proves (Prop. 1), and demonstrates the practical feasibility of (§ 4) the near-deterministic regime. To the best of our knowledge, all previous work relied on the linear case [44] or a (linear) approximation and the evaluation of the ELBO *around a point* to show the inductive bias on the decoder Jacobian. However, our main result (Thm. 1) yields a nonlinear equation where the decoder Jacobian can be evaluated at *any point* and is equipped with a convergence bound. Moreover, the consistency of VAE estimation for identifiable models [33] requires guarantees on $q_\phi(z|x)$; our result helps proving these. Dai and Wipf [13] use a non-factorized Gaussian variational posterior and prove in their Thm. 2 (including the $\dim x = \dim z$ case) that in the deterministic limit their $\kappa$-simple VAE can fit perfectly arbitrary observed data (barring few assumptions), while the ELBO gap tends to zero. In contrast, we use a factorized variational posterior; this prevents the ELBO gap to vanish in the deterministic limit, except in the special case of a decoder mean in the IMA class fitting the data perfectly. Dai and Wipf [13] take the limit of $\gamma \to +\infty$ (here using $\gamma$ as the square root of the decoder precision and not the decoder variance as used in [13]) to relate encoder and decoder properties in this limit in their Thm. 5, similarly to Prop. 1. In contrast to our nonlinear analysis, this is derived when optimizing w.r.t. both encoder and decoder parameters, and with a non-factorized encoder assumption, leading to fundamentally different behavior of the solutions in the deterministic limit. The work done by Sliwa et al. [62], simultaneously to ours, showcases an extensive empirical study highlighting that the IMA contrast allows distinguishing true and spurious solutions for a broad range of cases and outperforms standard regularizers such as weight decay. We discuss extended connections to the literature in Appx. D and Appx. E.

**Covariance structure and IMA.** We have shown that specific choices for encoder and decoder covariances regularize the decoder Jacobian, such that closing the ELBO gap constrains the decoder to belong to the IMA class. Following our intuition (Fig. 1), assuming factorized $q_\phi(z|x)$ and isotropic $p_\theta(x|z)$, IMA holds only for the *decoder*; since in the other direction the pushforward of $q_\phi(z|x)$ through $f^\theta$ has covariance $\mathbf{J}_{f^\theta}(z)\Sigma^\phi_{z|x}\mathbf{J}_{f^\theta}(z)^T$, which cannot be used to make row orthogonality statements on $\mathbf{J}_{f^\theta}(z)$ in the general case. Additionally, we conjecture that assuming an isotropic encoder would constrain IMA to hold in both encoding and decoding directions (as both $\mathbf{J}_{f^\theta}(z)$ and $\mathbf{J}_{g^\theta}(x)$ need to be column-orthogonal), such that the resulting decoder mean is constrained to have orthogonal columns of equal norms, which is a defining property of conformal maps [4]. On the other hand, we conjecture that if the observation model is not isotropic, but the encoder model is, IMA would only tend to be enforced for the mean *encoder* Jacobian, converging to the inverse decoder mean in the deterministic limit.

**Implications for recovering the true latent factors using unsupervised VAEs.** Convergence of the ELBO to the IMA-regularized log-likelihood suggests that unsupervised VAEs may recover the true factors of variation according to current identifiability results of the IMA class [4]. This is based on the following reasoning: *If the ground truth generative model belongs to the IMA class, unsupervised learning of the model with an infinite capacity VAE will, in the deterministic limit, ensure a solution that perfectly fits the data and whose decoder mean is also in the IMA class (by joint optimization of both the likelihood and the regularization term). Identifiability of the IMA class implies that the VAE will learn the true decoder (up to acceptable ambiguities); then, since self-consistency guarantees that the encoder inverts the decoder, the encoder infers the ground truth generative factors associated to observations.* Although strict identifiability for all functions in the IMA class remains to be proven, three concurrent papers provide guarantees that go towards identifiability: Leemann et al. [41] proves identifiability for a subset of the IMA class in the context of concept discovery; Zheng et al. [70] shows identifiability of nonlinear ICA by assuming a specific sparsity structure of the decoder Jacobian (called *structural sparsity*); whereas Buchholz et al. [4] introduce the concept of *local identifiability* and proves that IMA is locally identifiable.

Moreover, as mentioned in the above paragraph, we suspect that closing the ELBO gap with an isotropic encoder (while the encoder in Thm. 1 is only constrained to have diagonal covariance) constrains the decoder to be a conformal map. This is an interesting constraint, as nonlinear ICA with conformal mixings are identifiable: the two-dimensional case was first addressed with some additional constraints in [29], while the general case (in arbitrary dimension) was shown to rule out certain spurious solutions for conformal mixings [23], and finally proven to be identifiable by Buchholz et al. [4] in concurrent work. Hence, we conjecture that given a ground truth generative model with a conformal map from latent to observation space, and an unsupervised VAEs with isotropic Gaussian encoders and decoders, the true latent factors can be recovered.

**Conclusion.** We provide a theoretical justification for VAEs' widely-used self-consistency assumption in the near-deterministic regime of small decoder variance. Using this result, we show that the self-consistent ELBO converges to the IMA-regularized log-likelihood, and not to the unregularized one. Thus, we can characterize the gap between ELBO and true log-likelihood and reason about its role as an inductive bias for representation learning in nonlinear VAEs. We characterize a set of assumptions under which unsupervised VAEs can be expected to disentangle and we demonstrate this behavior in experiments on synthetic and image data.

## Acknowledgments and Disclosure of Funding

The authors thank the anonymous reviewers for their suggestions. This work was supported by the German Federal Ministry of Education and Research (BMBF): Tübingen AI Center, FKZ: 01IS18039A & 01IS18039B, and by the Machine Learning Cluster of Excellence, EXC number 2064/1 – Project number 390727645. Wieland Brendel acknowledges financial support via an Emmy Noether Grant funded by the German Research Foundation (DFG) under grant no. BR 6382/1-1. The authors thank the International Max Planck Research School for Intelligent Systems (IMPRS-IS) for supporting Dominik Zietlow and Patrik Reizinger. Patrik Reizinger acknowledges his membership in the European Laboratory for Learning and Intelligent Systems (ELLIS) PhD program.

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

## Acronyms

**ELBO** evidence lower bound
**IMA** Independent Mechanism Analysis

**i.i.d.** independent and identically distributed
**ICA** Independent Component Analysis

**KL** Kullback-Leibler Divergence

**LVM** Latent Variable Model

**MCC** Mean Correlation Coefficient

**MLE** Maximum Likelihood Estimation
**MLP** Multi-Layer Perceptron
**MSE** Mean Squared Error

**PCA** Principal Component Analysis
**PPCA** Probabilistic Principal Component Analysis

**SVD** Singular Value Decomposition

**VAE** Variational Autoencoder

## Nomenclature
**Independent Mechanism Analysis**
$C_{\mathbf{IMA}}$ global IMA contrast
$\alpha$ scalar field
$\mathbf{D}$ general diagonal matrix
$\mathbf{O}$ orthogonal matrix
$\boldsymbol{y}$ reconstructed sources
$\mathcal{L}_{\mathbf{IMA}}$ IMA loss function
$c_{\mathbf{IMA}}$ local IMA contrast

**Variational Autoencoder**
$\mathbf{V}$ weight matrix of a linear encoder
$\mathbf{W}$ weight matrix of a linear decoder
$\boldsymbol{\mu}^{\widehat{\phi}}(\boldsymbol{x})$ optimal mean of $q_\phi(\boldsymbol{z}|\boldsymbol{x})$
$\boldsymbol{\mu}^{\phi}(\boldsymbol{x})$ mean of $q_\phi(\boldsymbol{z}|\boldsymbol{x})$
$\phi$ parameters of the variational posterior $q_\phi(\boldsymbol{z}|\boldsymbol{x})$
$\boldsymbol{\sigma}^{\widehat{\phi}}(\boldsymbol{x})^2$ optimal variance of $q_\phi(\boldsymbol{z}|\boldsymbol{x})$
$\boldsymbol{\theta}$ parameters of the decoder $p_{\boldsymbol{\theta}}(\boldsymbol{x}|\boldsymbol{z})$
$\gamma$ square root of the precision of the VAE decoder
$\boldsymbol{\Sigma}^{\phi}_{\boldsymbol{z}|\boldsymbol{x}}$ covariance matrix of $q_\phi(\boldsymbol{z}|\boldsymbol{x})$
$\mathcal{L}_\beta$ $\beta$-VAE loss function
$\boldsymbol{f}^{\boldsymbol{\theta}}$ decoder
$\boldsymbol{g}^{\boldsymbol{\theta}}$ inverse decoder
$\widehat{\phi}$ optimal parameters of the variational posterior $q_\phi(\boldsymbol{z}|\boldsymbol{x})$
$p(\boldsymbol{x})$ data distribution
$p_0(\boldsymbol{z})$ latent prior distribution
$p_{\boldsymbol{\theta}}(\boldsymbol{z}|\boldsymbol{x})$ true posterior distribution of the decoded samples of the VAE, mapping $\boldsymbol{x} \mapsto \boldsymbol{z}$, parametrized by $\boldsymbol{\theta}$
$p_{\boldsymbol{\theta}}(\boldsymbol{x})$ marginal likelihood
$p_{\boldsymbol{\theta}}(\boldsymbol{x}|\boldsymbol{z})$ conditional distribution of the decoded samples of the VAE, mapping $\boldsymbol{z} \mapsto \boldsymbol{x}$, parametrized by $\boldsymbol{\theta}$

$q_\phi(z|x)$ variational posterior of the VAE, mapping $x \mapsto z$ parametrized by $\phi$

$q_{\widehat{\phi}}(z|x)$ optimal variational posterior of the VAE, mapping $x \mapsto z$ parametrized by $\phi$

$\mu_k^{\widehat{\phi}}(x)$ optimal mean of $q_\phi(z|x)$ in dimension $k$

$\mu_k^{\phi}(x)$ mean of $q_\phi(z|x)$ in dimension $k$

$\sigma_k^{\widehat{\phi}}(x)^2$ optimal variance of $q_\phi(z|x)$ in dimension $k$

$\sigma_k^{\phi}(x)^2$ variance of $q_\phi(z|x)$ in dimension $k$

$g^{\theta}$ inverse decoder component

$\mathbf{H}$ Hessian matrix

$\mathbf{I}_d$ $d$-dimensional identity matrix

$\mathbf{J}$ Jacobian matrix

$\boldsymbol{\Sigma}$ covariance matrix

$x$ observation vector

$z$ latent vector

$\mathcal{X}$ observation space

$d$ dimensionality of the observation space $\mathcal{X}$

$z$ latent single component

