# OpenReview forum: "Embrace the Gap: VAEs Perform Independent Mechanism Analysis"
_NeurIPS.cc/2022/Conference — NeurIPS 2022 Accept_

### Official Review · Reviewer_8jNM · 2022-07-07

**Rating:** 7
**Confidence:** 3
**Soundness:** 4 excellent
**Presentation:** 3 good
**Contribution:** 3 good

**Summary:**

The paper asks why VAEs learn useful representations despite the non-identifiability of the latent variables (in a nonlinear data generating process) using unregularized maximum likelihood estimation. The paper's main contribution is to show that VAEs with near-deterministic decoders do not converge to the marginal likelihood but to an IMA-regularized likelihood. The IMA-regularization (based on previous work) enforces a column-orthogonality condition on the Jacobian of the decoder, which can be beneficial for recovering the true latent factors when the data generating process satisfies the IMA assumption. Experimental results on synthetic numerical data corroborate that

- the encoder approximately inverts the decoder;
- the ELBO converges to the IMA-regularized likelihood if the true decoder is used, even when the generating process violates IMA;
- the encoder recovers the true latent factors if the data generating process satisfies the IMA assumption. The recovery is further corroborated by experiments on synthetic image data (Sprites).

The analysis focuses on near-deterministic decoders, for which the paper shows that the optimal encoder approximately inverts the decoder. This result is interesting in its own right as it formalizes an existing hypothesis and is necessary for deriving the convergence of the ELBO to the IMA-regularized likelihood.

**Questions:**

The abstract calls it a paradox that VAEs seem to learn "useful" representations despite not inverting the data generating process. As the usefulness of a representation depends on the downstream task, one could argue that a representation can be useful, even if it does not invert the data generating process. Maybe you could clarify this part in the abstract?

You assume that the variational posterior has a diagonal covariance matrix by design (e.g., in Figure 1 and Section 3.2).  However, is it safe to assume that the _empirical_ covariance is diagonal? Would a non-diagonal empirical covariance affect the IMA regularization? Similarly, what if the empirical covariance of the decoder is not isotropic?

In Definition 1, did you mean "mean decoder" instead of "mean encoder"?  In Definition 1 and Assumption 1, you assume that the mean decoder $f^{\theta}(z)$ is invertible. Is this assumption justified? I find it hard to imagine that the assumption is fulfilled at initialization time, even if the number of latent dimensions is equal to the number of input dimensions. But maybe there is an argument to be made for a model that reconstructs the inputs sufficiently well?

In section 4.1, to test the convergence of $\mu^{\hat{\phi}}(x)$ to $g^{\theta}(x)$, why do you not measure the MCC? Why is the comparison in the right subplot of Figure 2 more suitable?

For the numerical experiments you use different types of weight matrices (orthogonal, square upper-triangular, Möbius transform). Could you please clarify if the choice of weight matrix influences whether the mixing is in the IMA class?

Minor points:

- For the experiment in Section 4.1, you say that you use a uniform prior, but Table 1 lists a Gaussian prior.
- In line 79, you introduce the subscript $k$, but do not explain its purpose.

**Limitations:**

Overall, the paper is very transparent about the limitations.

One point for which I would have appreciated a bit more detail is the distinction between $\beta$ and $\gamma$. It is not immediately clear why self-consistency is harder to prove for $\beta$-VAEs, especially since you write that $\beta$ and $\frac{1}{\gamma^2}$ play a similar role in the objective (Section A.3).

The second point which could be addressed is the practical significance of the results (see Strengths and Weaknesses).

**Strengths And Weaknesses:**

**Strengths**

- Formalization of the mechanism by which VAEs can learn disentangled representations and comprehensive theoretical analysis with a detailed explanation.
- Thorough empirical analysis with controlled synthetic experiments.
- Clear and transparent contextualization of previous work, positioning of the paper, and contribution statement.
- Overall, a solid contribution that advances the understanding of representation learning with VAEs.

**Weaknesses**

- The paper does not discuss the practical significance of their results sufficiently well. For example: How much IMA violation would be acceptable to still recover the true factors in more tangible settings (e.g., in more complex variants of the Sprites dataset)? Can practitioners control or validate the degree of IMA violation when their goal is to recover the true but unknown latent factors? Even though the paper presents basic research, more experiments in practical settings could help readers better understand the practical significance of the theoretical results.
- The paper is dense and has a lengthy appendix. Subjectively speaking, the paper might be easier to parse and understand if it focused more on one part of the story.

---

> ### Author Response · Authors · 2022-08-02
> **Response to Reviewer 8jNM - Part 1/3**
>
> We thank Reviewer 8jNM for the suggestions and for thoroughly checking the technical details of our theoretical and empirical contributions.
>
>
> ## Assessing IMA violation in practice
>
> **Reviewer 8jNM:** How much IMA violation would be acceptable to still recover the true factors in more tangible settings (e.g., in more complex variants of the Sprites dataset)?
> **Reviewer 8jNM:** Can practitioners control or validate the degree of IMA violation when their goal is to recover the true but unknown latent factors?
>
> **Answer:** Precisely characterizing the violation of IMA in the context of the ground-truth generative process of real-world image datasets is nontrivial, since it would require computing the Jacobian of the ground truth mixing, something we typically do not have access to. An analogous issue arises in linear ICA, where the degree of non-Gaussianity required for identifiability from finitely many samples cannot be easily quantified in practice. For the image dataset which we consider in Section 4.3, we expect that the IMA principle holds approximately but not exactly. Nevertheless, our results suggest that VAEs are still able to recover the ground-truth sources. On the other hand, we point the reviewer to our Figure 5 left and to the experiments in Section 5.1 of (Zietlow et al., 2021), which suggest that severely perturbing the local structure of popular disentanglement data sets can lead to a significant drop in disentanglement performance of VAEs. Thus, our experiments and prior work suggest that, in image data sets, some degree of violation of the IMA-principle still allows for recovery of the ground-truth latents, however, as larger degrees of violation are introduced the model struggles to recover the ground-truth latents.
>
>
>
>
> ## Model assumptions
>
> **Reviewer 8jNM:** *You assume that the variational posterior has a diagonal covariance matrix by design (e.g., in Figure 1 and Section 3.2). However, is it safe to assume that the empirical covariance is diagonal?*
>
> **Answer:** We would like to start by expressing the possibility that we might have misunderstood what the reviewer meant by *empirical* covariance; namely, we are not aware of any definition of an empirical covariance for the variational posterior and how to measure it: in short, the variational posterior is constrained to a pre-specified model class that makes inference tractable. As a consequence, this posterior distribution will always be factorized irrespective of the distribution of the observations. We _kindly ask the reviewer to point us to references_ defining such empirical covariance.
>
>
> If the notion of empirical covariance relates to *covariances calculated from the (observed) data*, then we can state that our theory works independently of empirical quantities, since our proof only leverages the structural assumptions of the model, i.e., **our work does not impose any assumption on the empirical covariances** and more generally, VAE model assumptions are independent from statements on the data distribution.
>
> **Reviewer 8jNM:** *Would a non-diagonal empirical covariance affect the IMA regularization?*
>
> As we are not aware of the definition of such covariance in this context, so we will address the question for the encoder conditional covariance instead (here we thus mean the structural assumption, see the response to the previous point). A non-diagonal covariance of the encoder would imply a different structure of the Jacobian as pointed out in (Kumar and Poole, 2020). For example, a block-diagonal structure of the encoder covariance implies a block diagonal decoder Jacobian. Nonetheless, this case would violate our assumption of independent latent factors, which is generally assumed in the disentanglement/identifiablity literature.
>
> **Reviewer 8jNM:** *Similarly, what if the empirical covariance of the decoder is not isotropic?*
>
> **Answer:** If the decoder covariance is not isotropic (here we again mean the structural assumption as we are not aware of the definition of the empirical covariance of the decoder and kindly ask the reviewer to point us to the definition), then our theory is not applicable. On an intuitive level (as stated in Section 3.2, from line 187 in our original submission), an isotropic covariance is required to express the constraint that $J\_{f}^TJ\_{f}$ needs to be diagonal.

---

> > ### Author Response · Authors · 2022-08-02
> > **Response to Reviewer 8jNM - Part 2/3**
> >
> >
> >
> > **Reviewer 8jNM:** *In Definition 1 and Assumption 1, you assume that the mean decoder is invertible. Is this assumption justified? I find it hard to imagine that the assumption is fulfilled at initialization time, even if the number of latent dimensions is equal to the number of input dimensions. But maybe there is an argument to be made for a model that reconstructs the inputs sufficiently well?*
> >
> > **Answer:** As we mentioned, a key objective of our work is to draw a strong link between VAEs and exact likelihood optimization as performed **in the normalizing flow literature** on the one hand, **and nonlinear ICA literature** on the other hand. In both of these fields, **neural networks which are invertible at initialization are standard**. For example, (23) describes a multi-layer perceptron architecture which is invertible at initialization with probability one: when weight matrices are square, and nonlinearities invertible (e.g., smooth leaky ReLU or leaky tanh, see (23), Appendix H.3), then the whole neural network will be invertible with probability one under standard initialization scheme, for which the weight matrices will be invertible with probability one (e.g., individual entries sampled independently from a normal distribution). In the context of convolutional neural networks, see also (Ma et al., 2018), (Jacobsen et al., 2018), (Finzi et al., 2019). In short, **the convolution operation is invertible under mild conditions: Gaussian (or Uniform) sampled parameter tensors will yield invertible convolutional layers with probability one** (Ma et al., 2018), (Finzi et al., 2019). Moreover, the standard pooling operation can be replaced by an invertible operation (Jacobsen et al., 2018).
> >
> >
> > For the case where the (decoder) network maps from lower to higher dimensionality, we need to consider injective maps. In the context of ReLU Networks, this was characterized, for example, in (Paleka, 2021).
> >
> > Besides invertibility at initialization, we need to make sure that the _network stays invertible throughout training_. In likelihood maximization, **if the network is not invertible, then the Jacobian is not full rank and the log determinant of the decoder Jacobian term explodes** (goes to minus infinity). Maximizing the likelihood therefore encourages the network to stay invertible. Note that **our Theorem 1, Eq. (13) proves an analogy between $ELBO$ optimization and (regularized) maximum likelihood**, therefore we expect that invertibility is encouraged throughout training.
> >
> >
> >
> >
> > ## Weight matrix structures and the IMA class
> >
> > **Reviewer 8jNM:** For the numerical experiments you use different types of weight matrices (orthogonal, square upper-triangular, Möbius transform). Could you please clarify if the choice of weight matrix influences whether the mixing is in the IMA class?
> >
> > **Answer:** **We added a section to Appendix F (F.1) elaborating the reasons for choosing specific weight matrices.** In summary, the structure, functional map (nonlinearities/Möbius transform), and depth jointly determine whether the mixing is in the IMA class or not. A single-layer MLP with orthogonal weights is within, but a multilayer one is outside the IMA class; whereas the Möbius transform is always within. When we used triangular weights, we did so to ensure that the mixing is outside the IMA class (See Gresele et al., 2021, Appendix C.4, Lemma C.1).
> >
> >
> > ## The relationship $\beta$ and $\gamma$ for self-consistency (Proposition 1)
> >
> > **Reviewer 8jNM:** One point for which I would have appreciated a bit more detail is the distinction between $\beta$ and $\gamma$. It is not immediately clear why self-consistency is harder to prove for $\beta$-VAEs, especially since you write that $\beta$ and $\gamma$ play a similar role in the objective (Section A.3).
> >
> > **Answer:**
> > We thank the reviewer for this remark that pushed us to further investigate this question. We were initially concerned by the absence of a ``likelihood + gap'' decomposition similar to eq. (2) for the $\beta$-VAE objective, as this decomposition was central to our self-consistency result of Proposition 1. However, the reviewer is correct to point out, for $\gamma$ and $\beta$ fixed, both objectives are formally equivalent up to an affine rescaling that does not modify the localization of the optima. Indeed, we now show in Appendix A.3 that the obectives differ by a multiplication by a strictly positive constant and some additive constants. One caveat, we consider the asymptotic behavior of the $\beta$-VAE in the deterministic decoder limit, is that the constants involved diverge in this regime (see Eq. (19)). As consequence, it is possible to rescale the $\beta$-VAE objective such that it also converges to the IMA-regularized likelihood in the setting $\frac{\gamma}{\sqrt{\beta}}\to +\infty$, as claimed in Proposition 2 of appendix A.3. Whether this theory describes appropriately the common practice for $\beta$-VAE remains an open question.

---

> > > ### Author Response · Authors · 2022-08-02
> > > **Response to Reviewer 8jNM - Part 3/3**
> > >
> > > ## Clarifications
> > >
> > > **Reviewer 8jNM:** In Definition 1, did you mean "mean decoder" instead of "mean encoder"?
> > >
> > > **Answer:** We thank the reviewer for pointing out this discrepancy and changed the wording to mean decoder.
> > >
> > >
> > > **Reviewer 8jNM:** In Section 4.1, to test the convergence of $\mu^{\hat{\phi}}(x)$ to $g^{\theta}(x)$, why do you not measure the MCC? Why is the comparison in the right subplot of Figure 2 more suitable?
> > >
> > >
> > > **Answer:** The reason we plot the (log of) the squared distance between observations and the decoded mean encodings is that both Proposition 1 (self-consistency)  and Theorem 1 states its results in terms of a squared error. As we do not necessarily have access to the inverse of the decoder, we use the observations and the decoder, not the latents and the inverse decoder. That being said, MCC would indeed inform us about the quality of learning the true latents; however, **this experiment was designed to demonstrate a claim about the mean encoder and decoder, not about latent reconstruction in the sense of disentanglement/identifiability**; thus, we did not use MCC. On the other hand, the _usual_ MCC (with the Hungarian algorithm) is applied in a different context, where what matters is inversion of a ground truth function up to permutation/scale. In contrast, here our goal is to assess convergence of a learned function (the mean encoder) to the inverse of another learned function (the mean decoder), not up to the permutation/scale ambiguities.
> > >
> > > **Reviewer 8jNM:** For the experiment in Section 4.1, you say that you use a uniform prior, but Table 1 lists a Gaussian prior.
> > >
> > > **Answer:** We thank the reviewer for pointing out this discrepancy and changed the main text as we used a Gaussian prior.
> > >
> > > **Reviewer 8jNM:** In line 79, you introduce the subscript $k$, but do not explain its purpose.
> > >
> > > **Answer:** We thank the reviewer for pointing out this discrepancy and *added an explanation to line 80* (i.e., that the **subscript $k$ indexes the components of the factorized Gaussian of the encoder**)
> > >
> > >
> > > **Reviewer 8jNM:** The abstract calls it a paradox that VAEs seem to learn "useful" representations despite not inverting the data generating process. As the usefulness of a representation depends on the downstream task, one could argue that a representation can be useful, even if it does not invert the data generating process. Maybe you could clarify this part in the abstract?
> > >
> > > **Answer:** We acknowledge the reviewer's point that representations can indeed be useful for a downstream task even if the model does not invert the data generating process---this depends on the downstream task and the available model capacity. Disentangled representations can still be advantageous in that they tend to be (i) easier to use, and (ii) more data-efficient. Consider for example the task of predicting a given ground-truth factor (e.g., size) from the learned representation using a low-capacity function that can only select a single latent dimension: this will only be possible if size has been actually been disentangled/identified correctly. On the other hand, for a task like reconstruction with arbitrary capacity and infinite data, disentanglement is indeed not needed (or useful) in any meaningful way.
> > >
> > > Thus, **we changed the problematic sentence in the abstract to the following** (cfr. updated submission):
> > >
> > > _``While VAEs are commonly used for disentangled representation learning, it is unclear why $ELBO$ maximization would yield such representations, since unregularized maximum likelihood estimation generally cannot invert the data-generating process without additional assumptions."_
> > >
> > >
> > >
> > >
> > > ## References
> > > - (Zietlow et al., 2021) Demystifying Inductive Biases for $\beta$-VAE Based Architectures, in ICML
> > > - (Alyani et al., 2017): Diagonality measures of Hermitian positive-definite matrices with application to the approximate joint diagonalization problem, in Linear Algebra and its Applications
> > > - (Kumar and Poole, 2020): On Implicit Regularization in $\beta$-VAEs, in ICML
> > > - (Paleka, 2021) Injectivity of ReLU Neural Networks at Initialization
> > > - (Ma et al., 2018): Invertibility of convolutional generative networks from partial measurements, in NeurIPS
> > > - (Jacobsen et al., 2018): i-RevNet: Deep Invertible Networks, in ICLR
> > > - (Finzi et al., 2019): Invertible convolutional networks, in ICML (Workshop on Invertible Neural Nets and Normalizing Flows)

---

> > > > ### Comment · Reviewer_8jNM · 2022-08-08
> > > > **Response to the rebuttal**
> > > >
> > > > Thank you for the detailed response.
> > > >
> > > > Regarding the empirical covariance: yes, I meant the covariance calculated from the observed data. I agree that the variational posterior is factorized by assumption and that the paper's theoretical results only depend on the model assumptions and not on the empirical quantities. However, I wonder if the empirical covariance might still be non-diagonal (e.g., compare Figure 1 from [1]) and what this might imply for the encoder Jacobian and the IMA regularization, as it could suggest some form of model misspecification.
> > > >
> > > > Again, thanks for the clarifications and the adaptation of the manuscript. I will keep my score.
> > > >
> > > > **References**
> > > >
> > > > [1] Locatello, F., Bauer, S., Lucic, M., Raetsch, G., Gelly, S., Schölkopf, B., & Bachem, O. (2019). Challenging common assumptions in the unsupervised learning of disentangled representations. In ICML.

---

> > > > > ### Author Response · Authors · 2022-08-09
> > > > > **Response to the comments of Reviewer 8jNM**
> > > > >
> > > > > We thank **Reviewer 8jNM** for the discussion and the additional comments regarding the empirical covariances.
> > > > >
> > > > > ## Empirical covariance
> > > > >
> > > > > **Reviewer 8jNM:** _Regarding the empirical covariance: yes, I meant the covariance calculated from the observed data. I agree that the variational posterior is factorized by assumption and that the paper's theoretical results only depend on the model assumptions and not on the empirical quantities. However, I wonder if the empirical covariance might still be non-diagonal (e.g., compare Figure 1 from (Locatello et al., 2019)) and what this might imply for the encoder Jacobian and the IMA regularization, as it could suggest some form of model misspecification._
> > > > >
> > > > > **Answer:** If at convergence the empirical covariance is non-diagonal, then it may be that
> > > > >
> > > > > - (i) the true data generating process satisfies the IMA principle, but the model did not converge to a global optimum, or
> > > > > - (ii) the true data generating process does not satisfy the IMA assumptions (either non-independent latents or non-orthogonal Jacobian columns).
> > > > >
> > > > > For (i): To compare our results to the analysis of Figure 1 from (Locatello et al., 2019), we calculated the empirical covariance of the aggregate posterior for the Sprites experiments, for which we argue in the paper that the IMA principle may be approximately satisfied. Both the *sampled* and *mean* covariances are approximately diagonal. The non-diagonal entries are two or three orders of magnitude smaller than the diagonal ones, suggesting (together with the high MCC scores) that the recovered solution may, in fact, be close to optimal.
> > > > >
> > > > > For (ii): Under model misspecification of the latter kind (i.e., the true mixing's Jacobian has non-orthogonal columns), the question becomes to what extent is IMA regularization beneficial. Recent work (Sliwa et al., 2022) seems to suggest that using the IMA objective is still beneficial for learning the true latents even under some degree of violation of the IMA principle.
> > > > >
> > > > >
> > > > > ## References
> > > > > - (Locatello et al., 2019): Challenging common assumptions in the unsupervised learning of disentangled representations, in ICML.
> > > > > - (Sliwa et al., 2022): Probing the Robustness of Independent Mechanism Analysis for Representation Learning, in CRL at UAI.

---

### Official Review · Reviewer_UUBT · 2022-07-11

**Rating:** 6
**Confidence:** 4
**Soundness:** 3 good
**Presentation:** 2 fair
**Contribution:** 2 fair

**Summary:**

This work proves that for Guassian VAEs with isotropic Gaussian encoders, when the support of the data $\mathbf{x}\in\mathbb{R}^d$ has full dimensionality $d$,

- The ELBO optima has decoder variance tend to zero ("self-consistency", Sec 3.1), and that
- ELBO converges to a regularized likelihood objective where the regularization encourages the decoder Jacobian to have orthogonoal columns (Sec 3.2).

The latter is connected to the recent mechanism independence formulation for causal representation learning, and the authors provide experiments demonstrating the recovery of the true latents when the mechanism independence assumptions are met.

**Questions:**

I think the paper should be revised to better position the existing results, or to improve them in the aforementioned directions.  Some of the discussions in Appendix D could also be moved to the main text.

**Limitations:**

Some limitations, e.g., the dimensionality issue, are mentioned, but their implications not fully discussed (see above).  Others are adequately discussed.

**Strengths And Weaknesses:**

The established connection of ELBO-trained VAEs to mechanism independence is interesting, and the experimental results can be of value to the causal representation learning community (I'm not deeply familiar with that literature).  The theoretical results are new, in the sense that they have not been established in this exact setting.

That being said, I feel that the significance of the results, in the broader context of understanding representation learning in ELBO trained VAE, is overstated.  It has been known that

- Linear-Gaussian VAE trained with ELBO recover the principal components, and thus their decoder (Jacobian) have orthogonal columns ([4, 60]; Dai et al, 2018);
- Nonlinear Gaussian VAE trained with ELBO are regularized towards having decoders with orthogonoal Jacobian columns ([43, Section 5, 5.1]; Nakagawa et al, 2021);

Dai and Wipf (2019) has also shown that near-deterministic VAEs constitute a family of global optima of the Gaussian VAE ELBO, and empirically verified that it is recovered by common VAE architectures.

It is true that no results have been provided in the precise setting of this work, but I think this has more to do with the restrictiveness of the setting here: this work assumes the data distribution has full support (by assuming the optimal ELBO is finite), and it also uses an isotropic encoder with output dimensionality $d_z = \mathrm{dim}\\;\mathbf{x}$. The former is inappropriate for image or other structured inputs which are typically supported on low-dimensional manifolds embedded in $\mathbb{R}^{d_x}$, while the latter may prevent the automatic adaptation to the correct latent dimensionality (Dai and Wipf, 2019).  Relaxation of either restriction introduces technical challenges, so I don't think readers familiar with previous work on VAE inductive bias will find the results here surprising.  The related works are only sparingly discussed in the main text, which does not help to put the contributions of this work in context.

I would also note that self-consistency may cease to hold beyond the $d_z = \mathrm{dim}\\:\mathbf{x}$ regime.  Suppose for simplicity that $\mathrm{dim}\\:\mathbf{x}=1$.  Then any input $x$ can be encoded up to precision $O(2^{-d_z})$ by setting $q(z_i | x) = \mathcal{N}(C s_i(x), 1)$, where $C>1$ is a moderate constant, and $s_i(x)\in\\{1, -1\\}$. Then we have $\mathrm{sgn}(z_i) = s_i(x)$ with high probability, for $z\sim q(z|x)$, and the sign decoder can be easily approximated with NN decoders.  More generally, we can always take an input dimensionality $i\in [\mathrm{dim}\\:\mathbf{x}]$ and approximate it with precision $\gamma$, using up to $O(\log \gamma)$ latent dimensions which are *never deterministic in the limit*.  While this issue does not prevent the use of ELBO-trained VAEs for the recovery of independent mechanisms, thus the contribution of this work in this context, it does make it less interesting in the broader context of understanding VAEs (which are often overparamterized in a similar nature).

## References

- Nakagawa et al (2021): Quantitative Understanding of VAE as a Non-linearly Scaled Isometric Embedding, in ICML.
- Dai and Wipf (2019): Diagnosing and Enhancing VAE Models, in ICLR.
- Dai et al (2018): Connections with Robust PCA and the Role of Emergent Sparsity in Variational Autoencoder Models, in JMLR.

---

> ### Author Response · Authors · 2022-08-02
> **Response to Reviewer UUBT - Part 1/3**
>
> We thank Reviewer UUBT for pointing us to relevant references we did not cite in the original submission.
>
> ## Related work
>
> **Reviewer UUBT:** *the significance of the results, in the broader context of understanding representation learning in $ELBO$ trained VAE, is overstated.
> It has been known that:
> (i) Linear-Gaussian VAE trained with $ELBO$ recover the principal components ([4, 60]; Dai et al., 2018);
> (ii) Gaussian VAE trained with $ELBO$ are regularized towards having decoders with orthogonal Jacobian columns ([43, Section 5, 5.1); (Nakagawa et al., 2021)
> (Dai and Wipf, 2019) have also shown that near-deterministic VAEs constitute a family of global optima of the Gaussian VAE $ELBO$, and empirically verified that it is recovered by common VAE architectures.*
>
> **Answer:**  Firstly, thanks for mentioning (Dai et al, 2018), (Nakagawa et al, 2021) and (Dai and Wipf, 2019), which we regretfully left out of our list of references. **We added them** in the revised version of the paper, see **Appendix D** (citations will be reintegrated to the 10 pages main text of the camera ready version).
>
> We are aware of **(i)**: in fact, and as we acknowledge at multiple points in the paper, the results of [4, 60], among others, are key references we built upon. However, and crucially, these results are **linear** whereas our result is **nonlinear**: While the analyses in [4, 60] require linearization of the objective around a single data point and (Dai et al., 2018) uses affine functions, **our work presents a rigorous analysis of the **nonlinear** $ELBO$ objective in the near-deterministic regime.**
>
> Regarding **(ii)**: One of the references mentioned by the reviewer (we presume the reviewer meant [42, Section 5, 5.1], not [43, Section 5, 5.1]) does not characterize how the $ELBO$ behaves as $\gamma$ goes to infinity. In fact, the authors of [42] clearly state that their analysis in Section 5.1 is based on a “deterministic approximation” of the $ELBO$ objective they introduce in Section 4. For the $ELBO$ itself, the relationship with the exact likelihood was unclear prior to our work: For example, another prominent reference [52] stated that for $\gamma=\infty$ the (i.e., “in” the deterministic limit) the $ELBO$ equals the exact likelihood one would find through a change of variables, and notably without regularization. In contrast, **our analysis shows that as $\gamma$ goes to infinity the gap does not vanish in general, and can be characterized through the global IMA contrast.**
>
> Thank you for pointing us to (Dai and Wipf, 2019), which used a different setting that turns out to be very interesting to compare to ours. The most important is: while **we use a factorized Gaussian variational posterior, (Dai and Wipf, 2019) use a non-factorized Gaussian**, which leads to major differences. Broadly construed, (Dai and Wipf, 2019) are able to show in their Theorem 2 (which includes the case of equal latent and observation dimensions matching our setting) that in the deterministic limit, their $\kappa$-simple VAE can fit perfectly the arbitrary observed data (barring few assumptions), while the $ELBO$ gap tends to zero. The way it is proven takes a first step, where they rely on the "Darmois construction" introduced in (Hyvärinen et al., 1999) to choose the decoder mean parameter such that its pushforward is exactly the observational distribution, up to convolution by the decoder's isotropic Gaussian kernel (which can be ignored in the deterministic limit). Then in a second step, with an appropriate choice of variational posterior parameters, they show that asymptotically the $ELBO$ gap (i.e., the KL divergence between true and variational posterior) tends to zero in the deterministic limit. In contrast, our factorized variational posterior does not allow the $ELBO$ gap to vanish in the deterministic limit, unless of course the decoder mean is in the IMA class. For this reason, _the proofs and scope of our results are very different:_
>
> - **We use information theoretic bounds to show that the encoder inverts the decoder mean** (independently from the fact that this one may or may not fit the data perfectly);
> - **We obtain a rigorous convergence to the IMA regularized likelihood**, which demonstrates that **the gap is not eliminated in the deterministic limit**.
>
>
> Moreover, none of the references mentioned by the reviewer characterize the behavior of the $ELBO$ as $\gamma$ tends to infinity using “big O” notation, and crucially this does not allow them to make a statement as precise as the one in our Theorem 1, eq. (13)---which connects the $ELBO$ and a regularized log-likelihood. In short, **the analyses and tools presented in previous works were insufficient to characterize the non-vanishing gap of near deterministic VAEs with factorized encoders as we do.**

---

> > ### Author Response · Authors · 2022-08-02
> > **Response to Reviewer UUBT - Part 2/3**
> >
> >
> > An interesting question is then which variational posterior model is used in practice: factorized or not? We are only aware of works using a factorized encoder ( (Rolinek et al., 2019), (Kumar and Poole, 2020), (Lucas et al., 2019) ), most likely because a non-factorized one would be challenging to estimate as the latent dimension gets large. Our result shows that this choice of factorized posterior is fortunate as it has the additional benefit of favoring disentanglement. **We added clarification on this in our Appendix D, where we thoroughly discuss related work.**
> >
> >
> > **Reviewer UUBT:** The related works are only sparingly discussed in the main text, which does not help to put the contributions of this work in context.
> >
> > **Answer:** We start by noting that **Reviewer 8jNM** stated that our paper presents a **“clear and transparent contextualization of previous work, positioning of the paper, and contribution statement”**. Nevertheless, the literature on VAEs is extremely vast, and we missed the references discussed above. In the revised version of the paper we uploaded, **we added references to the papers the reviewer mentioned, see Appendix D.**
> >
> > At the same time, as expressed in the answer to one of the points above, we believe that the additional references suggested by the reviewer do not undermine the novelty or significance of our work. We also point to Appendix D in the supplemental, where we discussed many related works in detail, since the space in the main text is limited. For convenience, we provide a compressed version of the changes we made:
> >
> > - The paper by (Nakagawa et al., 2021) builds upon the results of (Rolinek et al., 2019) and provides a _novel interpretation_ of a VAE via introducing implicit variables as a latent variable model with isometric embeddings.
> >
> > - (Dai et al, 2018) provide a more general analysis (affine functions) than, e.g., (Lucas et al., 2019) (linear Gaussian VAE). The focus on Robust PCA also indicates a slightly different emphasis than our paper, although the concept of orthogonality is indeed present via PCA.''
> >
> > - The approach (Dai and Wipf, 2019) take on VAEs is  different according to our understanding: their main goal is to investigate the source of why VAEs fail to capture the true data distribution. In the sense of analyzing the optimal solution of ELBO optimization, the connection is very strong indeed. Their analysis also concerns the case when the latents have the same dimensionality as the observations, but they extend to the  $\dim x > \dim z$ case.
> >
> > ## Assumptions
> >
> > **Reviewer UUBT:** The setting is restrictive: (i) the data distribution needs to have full support (by assuming the optimal $ELBO$ is finite),
> >
> > **Answer:** We note that this same assumption is used in some prominent literature such as [52]. Technically, this assumption is required to draw the link to the usual change of variables used in, e.g., normalizing flows (see [53]). We believe that extending it to manifolds in higher dimensions is possible but it presents some technical difficulties discussed in more detail in Section 5 of our paper. Nevertheless, our empirical results in Section 4.3 suggest that a similar regularization towards Jacobians with orthogonal columns as the one predicted in our Thm. 1 takes place in the high-dimensional setting, where the observations lie on a manifold embedded in a space whose dimensionality is higher than the number of latent variables.

---

> > > ### Author Response · Authors · 2022-08-02
> > > **Response to Reviewer UUBT - Part 3/3**
> > >
> > >
> > > **Reviewer UUBT:** The setting is restrictive: (ii) it uses an isotropic encoder with output dimensionality $\dim z=\dim x$
> > >
> > >
> > > **Answer:** This misunderstanding might have been caused by the fact that we finished our paper with a conjecture about a special case of VAEs, namely one with isotropic encoder variance. In hindsight, we realize that we put this in a very prominent position in the discussion, which might suggest that this setting is central to our paper. But it is not, since throughout the paper (except Conjecture 1), **we consider for our main results encoders with diagonal covariance where each dimension can have different variances**. In the revised version of the paper, we made the distinction clearer by adding the clarification **"the encoder in Theorem 1 has diagonal covariance"**.
> > >
> > > We do assume an isotropic _decoder_. This is a quite standard assumption, and also exploited in other works mentioned by the reviewer, such as, e.g.,(Dai and Wipf, 2019). We can in principle also extend our treatment to non-isotropic decoders, but this would likely imply looser constraints on the decoder’s function class, which is one of the main points of our paper: Conditions on the distribution involved in the $ELBO$ variational objective implicitly constrain the decoder.
> > >
> > > That being said, our setup concerns the most widely-used VAE (Gaussian encoder with diagonal covariance, Gaussian decoder with isotropic covariance, Gaussian prior), so the main restriction of our theory is assuming same latent and observation dimensionalities. However, in practice, we show that there is evidence indicating that our results might hold when those dimensionalities are different (cf. the image experiments in Section 4.3). To address this practically very important scenario, an extension to IMA theory is required. We are optimistic since there is recent work inspired by the IMA paper (eg,(Cunningham et al., 2022) ) - when the extension is provided for IMA, we will also accommodate it into our theory.
> > >
> > >
> > >
> > >
> > > ## Self-consistency
> > >
> > > **Reviewer UUBT:** I would also note that self-consistency may cease to hold beyond the $\dim z = \dim x$ regime.
> > >
> > > **Answer:** Concerning the point w.r.t. the possible breakdown of self-consistency, we acknowledge that further analysis is required to develop a better understanding of the setting with dimensionality mismatch between latent sources and observations.
> > >
> > > However, the construction described by the reviewer seems to require an overcomplete representation (i.e., more latent sources than the observations), which is not our focus in this paper as it is not a setting common in representation learning.
> > >
> > >
> > >
> > > ## References
> > > - (Nakagawa et al., 2021): Quantitative Understanding of VAE as a Non-linearly Scaled Isometric Embedding, in ICML.
> > > - (Dai and Wipf, 2019): Diagnosing and Enhancing VAE Models, in ICLR.
> > > - (Dai et al., 2018): Connections with Robust PCA and the Role of Emergent Sparsity in Variational Autoencoder Models, in JMLR.
> > > - (Rolinek et al., 2019): Variational Autoencoders Pursue PCA Directions (by Accident), in CVPR
> > > - (Kumar and Poole, 2020): On Implicit Regularization in $\beta$-VAEs, in ICML
> > > - (Lucas et al., 2019): Don't Blame the ELBO! A Linear VAE Perspective on Posterior Collapse, in NeurIPS
> > > - (Cunningham et al., 2022): Principal Manifold Flows, arXiv

---

> > ### Comment · Reviewer_UUBT · 2022-08-07
> > **Reviewer's Response**
> >
> > Thank you for your response.  I am increasing my score to 6, because (i) my original claim that this work assumes isotropic encoder covariance was incorrect and (ii) the updated discussion on related work.
> >
> > I appreciate the authors' effort to improve the discussion of related work within the current page limit.  However, I strongly recommend the authors to utilize the extra page allowed in camera ready -- if this submission is accepted -- to better discuss past results on VAE ELBO analysis, especially because previous works provide solid intuition on the technical results established here.  To provide a truly transparent contextualization of previous literature and the (*technical* side of the) contributions here, such discussions should be ideally moved to the main text, for example, around the earlier references to the "self-consistency assumption".  This issue is separate from this work's contribution in relating the analyses to the study of mechanism independence, which is solid and well presented.
> >
> > While I do not have any objection against the acceptance of this work, my current rating reflects my belief that the restriction to the full-dimensional case ($\mathrm{dim} \mathbf{x} = d_z$) is non-trivial, and limits the broader impact of this work, as well as all past works with similar restrictions.

---

> > > ### Author Response · Authors · 2022-08-09
> > > **Response to the comments of Reviewer UUBT**
> > >
> > > We thank **Reviewer UUBT** for taking our response into consideration in the revised evaluation. **We will definitely include the discussion of the related work in the main text of the camera-ready version**, incorporating the papers pointed out by the reviewer and discussing their relationship to the results presented in our paper.
> > >
> > > ## Higher-dimensional observations
> > >
> > > **Reviewer UUBT:** _...my current rating reflects my belief that the restriction to the full-dimensional case $(\dim x \neq \dim z)$ is non-trivial, and limits the broader impact of this work, as well as all past works with similar restrictions._
> > >
> > > **Answer:** While our current work does not discuss the $\dim x \neq \dim z$ case, we believe that the connection between the $ELBO$ and the IMA-regularized log-likelihood extends to the higher-dimensional observation case. Our empirical results on the Sprites dataset also suggest this since we found that the Jacobian columns for the learned decoder are approximately orthogonal. Theoretical characterization of this is, however, left for future work, where we believe we can draw inspiration from (Dai and Wipf, 2019), as the authors provide an interesting framework to investigate such a setting.
> > >
> > > Another topic for future investigation is the identifiability of IMA when $\dim x \neq \dim z$. Existing identifiability proofs for other ICA settings (e.g., Khemakhem et al., 2020); or, in the context of concept discovery, for a subset of the IMA class (Leemann et al., 2022)) may possibly provide a starting point.
> > > In this regard, we note that some identifiability results for nonlinear ICA, originally formulated for the case where $\dim x = \dim z$ (Hyvärinen et al., 2019), were later also extended to the setting where $\dim x > \dim z$ (Khemakhem et al., 2020), suggesting that a similar extension could be possible.
> > >
> > >
> > >
> > > ## References
> > > - (Dai and Wipf, 2019): Diagnosing and Enhancing VAE Models, in ICLR.
> > > - (Hyvärinen et al., 2019): Nonlinear ICA Using Auxiliary Variables and Generalized Contrastive Learning, in AISTATS.
> > > - (Khemakhem et al., 2020): Variational Autoencoders and Nonlinear ICA: A Unifying Framework, in AISTATS.
> > > - (Leemann et al., 2022): Disentangling Embedding Spaces with Minimal Distributional Assumptions, in arXiv.

---

### Official Review · Reviewer_tFG1 · 2022-07-12

**Rating:** 6
**Confidence:** 2
**Soundness:** 3 good
**Presentation:** 3 good
**Contribution:** 3 good

**Summary:**

The paper analyzes the gap between ELBO and exact log-likelihood under the self-consistency assumption that the encoder is the maximizer of ELBO with the decoder parameters being fixed. The authors theoretically show that the gap can be characterized with independent mechanism analysis (IMA) in a near-deterministic regime. Experiments empirically support the theoretical results.

**Questions:**

- What does the limitation that dimensionality of $x$ and $z$ should be the same come from? It is unclear to the reviewer whether $c_\text{IMA}$ in Section 4.3 is valid for the theory part.

**Limitations:**

Limitations are addressed, and no negative societal impact is observed.

**Strengths And Weaknesses:**

### Strong points
Understanding the gap between ELBO and exact log-likelihood is important. The analysis/interpretation of the gap from the perspective of IMA is new and interesting. The analysis provides new insight into disentanglement in (usual) VAE. In addition, the empirical results successfully support the theoretical parts.

### Weak points
- In the experiments, $\gamma$ is always fixed. Since $\gamma$ is optimized as well as encoder and decoder in the current VAEs, I would like to have seen a case with trainable $\gamma$.
- How large the gap between ELBO optimized in the usual way and ELBO^* is not clear.

---

> ### Author Response · Authors · 2022-08-02
> **Response to Reviewer tFG1 - Part 1/2**
>
> We thank Reviewer tFG1 for the valuable feedback and for pointing out practically and theoretically relevant questions about our submission. In the following, we address the remarks point-by-point.
>
> ## Trainable $\gamma$
>
> **Reviewer tFG1:** *In the experiments, $\gamma$ is always fixed. Since $\gamma$  is optimized as well as encoder and decoder in the current VAEs, I would like to have seen a case with trainable $\gamma$*.
>
> **Answer:** The reviewer raised an important point: we did not investigate the case of trainable $\gamma$ in our original submission, despite the fact that tuning the decoder variance can lead to superior sample quality, as noted by, e.g., (Rybkin et al., 2021). Unfortunately, as acknowledged by the same authors, *optimizing the decoder variance can be hard*, as it requires a careful tuning of the learning rate and leads to a suboptimal likelihood initially---this is the reason the literature generally omits learning the decoder variance (see, e.g., (Rolinek et al., 2019), (Kumar and Poole, 2020), (Lucas et al., 2019) ).  In accordance to the reviewer's request, we reran experiments from Section 4.3 of our paper with a trainable decoder variance (everything else being unchanged)---hereby we confirm the observation of (Rybkin et al., 2021); namely, that optimization becomes much harder, leading to suboptimal results in terms of MCC. To achieve superior results, one would possibly need to do extensive hyperparameter tuning. Possibly, the implementation for the $ELBO$ also needs to change to accomodate a trainable $\gamma$---cf. the paragraph *Loss implementation details* in Section 3.1 of (Rybkin et al., 2021).
>
> Thus, to investigate the trainable $\gamma$ setting and compare it to our original experiments, we utilized the theoretical results of (Rybkin et al., 2021) to *analytically calculate the maximum likelihood estimate of the decoder variance*. As noted in Section 4 of that paper, this has the form of $MSE(x; f(\mu(x)))$, i.e., the mean squared error between the samples and the decoded *mean encodings*.
>
> We **added an analysis to Appendix F.5** and report the *comparison between the maximum likelihood estimate of $\gamma$ and our hyperparameter setting* in Fig. 7. We can observe that for all $\gamma^2$ below $1e5$, the corresponding $MSE$ is smaller by approximately one order of magnitude than the decoder variance (i.e., $1/\gamma^2$), indicating that the selected hyperparameter is potentially *too small*. However, for $\gamma=1e5,$ the $MSE$ values (for the 20 seeds) lie in the range $[0.8e5;3.8e5]$ with a mean and standard deviation of $2.3e5\pm 0.77e5,$ indicating that **our hyperparameter setting of $1e5$ for $\gamma$ and the maximum likelihood estimate are in the same order of magnitude**, corroborating the fact that we used the optimal setting (up to the granularity of our original grid search) from the values $1e1,1e2,1e3,1e4,1e5$.
>
> ## Characterizing the gap between $ELBO$ and $ELBO^{\*}$
>
> **Reviewer tFG1:** *How large [is] the gap between $ELBO$ optimized in the usual way and $ELBO^{\*}$?*
>
> **Answer:** The gap between $ELBO$ and $ELBO^{\*}$ for a generic setting of the encoder’s parameters (e.g., at initialization) is hard to characterize in theory. In particular, as there is no closed form solution for the optimal variational posterior in the nonlinear setting, we do not have a way to estimate the $ELBO^{\*}$.
> In practice, we see that **our theory for the $ELBO^{\*}$ is predictive for the outcomes of optimization of the $ELBO$**. For example, note that the experiments in Figure 2 and section 4.1 are performed optimizing the $ELBO$. This appears to indicate that the gap between $ELBO$ and $ELBO^{\*}$ vanishes during optimization for the cases we considered (i.e., the learned encoder is indeed almost optimal w.r.t. the decoder).
> The same is corroborated in Section 4.2 when comparing the $ELBO^{\*}$, the regularized and unregularized log-likelihoods: although we use the notation of $ELBO^{\*}$ in Figure 4, we calculate its value given the standard quantities of VAE training, i.e., practically it is the $ELBO$. Since the relation of Theorem 1 (Eq. 13) holds for the $ELBO$ as well, we can say that given a sufficiently expressive model, it is possible to achieve the optimum.

---

> > ### Author Response · Authors · 2022-08-02
> > **Response to Reviewer tFG1 - Part 2/2**
> >
> >
> > ## Theory for mismatching latent and observation dimensionality
> >
> > **Reviewer tFG1:** *Where does the limitation that dimensionality of latents and observations should be the same come from?*
> >
> > **Answer:** This limitation comes from the theory of IMA and (most) theoretical work investigating nonlinear ICA as well as normalizing flows (standard ones) (Hyvärinen and Morioka, 2016), (Papamakarios et al., 2021), (Gresele et al., 2021). Namely, it is only defined when latents and observations have the same dimensionality (which is the general setting in the identifiability literature). Although there is work to extend the concept to the practically relevant case of higher observation dimensionality (cf. (Cunningham et al., 2022) for applying the idea to normalizing flows), the theoretical properties of IMA in such setting have not yet been characterized.
> >
> > **Reviewer tFG1:** *It is unclear to the reviewer whether $C_{IMA}$ in Section 4.3 is valid for the theory part.*
> >
> > **Answer:** We address the reviewer’s remarks for each scenario we investigated in Section 4.3. We added an analysis of using $C_{IMA}$ in Appendix A.4 and summarize it in the following.
> >
> > - For the *synthetic data*, as the observation and latent dimensions are the same, $C\_{IMA}$ has the theoretical properties shown in (Gresele et al., 2021)
> > - For *image data*, the theoretical assumptions of (Gresele et al., 2021) are not met (the observation dimension is much larger than the latent dimension), meaning that we cannot say that the beneficial properties of IMA for identifiability rigorously apply for the $\dim x \neq \dim z$ case. Our goal with these experiments was to investigate whether our theory may hold in the setting with different latent and observation dimensionalities, which is standard in the context of representation learning.  However, **$C\_{IMA}$ is a valid metric on its own to measure the column-orthogonality of the decoder Jacobian even for image data** due to the following reasons:
> >     1. $C\_{IMA}$ of the decoder Jacobian $J\_{f}$ is the left KL measure of diagonality of $J\_{f}^TJ\_{f}$ (Alyani et al., 2017)---which is also pointed out in Eq. 23 in Appendix C.1 of (Gresele et al., 2021). Since this alternative formulation is defined in terms of $J\_{f}^TJ\_{f}$, which is *always a symmetric, square matrix*.
> >     2. We calculate $C\_{IMA}$ according to Eq. 23 in Appendix C.1 of (Gresele et al., 2021), meaning that **we can use the same formula for the $\dim x \neq \dim z$ case**; thus, the numerical values we report are not subject to further restrictions.
> > We acknowledge that we did not emphasize these facts in the original submission, and **we added Appendix A.4 to clarify this in the revised version.** The experimental results are promising: thus, we believe that there is potential for an extension of our theory to accommodate our observations in the image experiments.
> >
> > ## References
> > - (Rolinek et al., 2019): Variational Autoencoders Pursue PCA Directions (by Accident), in CVPR
> > - (Kumar and Poole, 2020): On Implicit Regularization in $\beta$-VAEs, in ICML
> > - (Lucas et al., 2019): Don't Blame the ELBO! A Linear VAE Perspective on Posterior Collapse, in NeurIPS
> > - (Rybkin et al., 2021): Simple and Effective VAE Training with Calibrated Decoders, in ICML
> > - (Seitzer et al., 2021): On the pitfalls of heteroscedastic uncertainty estimation with probabilistic neural networks, in ICLR
> > - (Cunningham et al., 2022): Principal Manifold Flows,
> > - (Alyani et al., 2017): Diagonality measures of Hermitian positive-definite matrices with application to the approximate joint diagonalization problem, in Linear Algebra and its Applications
> > - (Hyvarien et al., 1999): Nonlinear independent component analysis: Existence and uniqueness results, in Neural Networks
> > - (Papamakarios et al., 2021): Normalizing Flows for Probabilistic Modeling and Inference, in JMLR
> > - (Gresele et al., 2021): Independent mechanisms analysis, a new concept?, in NeurIPS
> > - (Hyvärinen and Morioka, 2016): Unsupervised Feature Extraction by Time-Contrastive Learning and Nonlinear ICA, in NeurIPS

---

> > ### Comment · Area_Chair_mHKk · 2022-08-09
> > **Reviewer tFG1: Has this response (in part) addressed your concerns?**
> >
> > _Reviewer tFG1_: Could you comment on the responses with respect to trainable $\gamma$ and characterizing the gap between the ELBO and ELBO*? Thanks!

---

### Author Response · Authors · 2022-08-02
**Response to all reviewers**

We thank all reviewers for their valuable feedback and thoughtful suggestions that improved our submission. We appreciate the reviewers' assessment of our work as _“important''_ for _“understanding the gap between $ELBO$ and exact log-likelihood”; “The analysis [...] of the gap from the perspective of IMA is new and interesting”_ **(Reviewer tFG1)**; _“The [...] connection of $ELBO$-trained VAEs to mechanism independence is interesting”_ **(Reviewer UUBT)**.

Reviewers also appreciated our _“formalization of the mechanism by which VAEs can learn disentangled representations and comprehensive theoretical analysis with a detailed explanation”_ **(Reviewer 8jNM)**; and that _“the analysis provides new insight into disentanglement in (usual) VAE”_ **(Reviewer tFG1)**; and emphasized our _“thorough empirical analysis”_ **(Reviewer 8jNM)**; _“the empirical results successfully support the theoretical parts”_ **(Reviewer tFG1)**; _“the experimental results can be of value [for] causal representation learning”_ **(Reviewer UUBT)**.

**Reviewers tFG1 and UUBT** questioned the assumption that **observations and latent variables should have the same dimensionality**. We emphasize that our main goal was to connect unsupervised VAEs to the theoretical results **in the nonlinear ICA and normalizing flows literature, where this assumption is standard,** and is similar to, e.g., (Nielsen et al., 2021, Appendix A) (see also the Limitations section of our submission). We acknowledge  that higher-dimensional observations are practically more relevant for representation learning. Therefore, we provided promising empirical evidence for image data (see Section 4.3): we show in Figure 5 (left) that $C\_{IMA}$ of the learned decoder and the $ELBO$ are anti-correlated during training---suggesting that a similar regularization as predicted in our Thm. 1 takes place in the high-dimensional setting.


While **Reviewer UUBT** doubts that _“readers familiar with previous work on VAE inductive bias will find the results here surprising”_, we note that other reviewers seem not to agree with this characterization of our work, stating that _“The analysis [...] of the gap from the perspective of IMA is new and interesting. [It] provides new insight into disentanglement in (usual) VAE”_ **(Reviewer tFG1)** and _“a solid contribution that advances the understanding of representation learning with VAEs”_ **(Reviewer 8jNM)**.

**Reviewer UUBT** also said that related work is  _“only sparingly discussed in the main text, which does not help to put the contributions of this work in context”_. However, **Reviewer 8jNM** did not share this view: _“clear and transparent contextualization of previous work, positioning of the paper, and contribution statement”_.
Nevertheless, we thank **Reviewer UUBT** for mentioning some related work we had not cited, but now **discuss in Appendix D,** and we will include it in the camera-ready version. In short, this related work provides interesting additional context, but we believe it does not undermine the novelty and significance of our results.


## References
- (Nakagawa et al., 2021): Quantitative Understanding of VAE as a Non-linearly Scaled Isometric Embedding, in ICML.
- (Dai and Wipf, 2019): Diagnosing and Enhancing VAE Models, in ICLR.
- (Dai et al., 2018): Connections with Robust PCA and the Role of Emergent Sparsity in Variational Autoencoder Models, in JMLR.
- (Nielsen et al., 2021): SurVAE Flows: Surjections to Bridge the Gap between VAEs and Flows, in NeurIPS

---

### Meta-Review · Area_Chair_mHKk · 2022-08-31

**Recommendation:** Accept
**Confidence:** Certain

**Metareview:**

## MetaReview

**Summary**: The submission examins why VAEs learn useful representations despite the non-identifiability of the latent variables (in a nonlinear data generating process). To do so they analyze the amortization gap from the perspective of independent mechanism analysis (IMA). The main contribution is to show that VAEs with near-deterministic decoders do not converge to the marginal likelihood but to an IMA-regularized likelihood. Previous work has shown that this regularization enforces a column-orthogonality condition on the Jacobian of the decoder, which aids in recovering the true latent factors when the data generating process satisfies the IMA assumption. The analysis focuses on near-deterministic decoders, for which the paper shows that the optimal encoder approximately inverts the decoder. This result  formalizes an existing hypothesis and is necessary for deriving the convergence of the ELBO to the IMA-regularized likelihood. Synthetic experiments empirically support theoretical results.

**Strengths**: Reviewers [tFG1] and [UUBT] appreciate the  analysis of the amortization gap from the perspective of independent mechanism analysis, finding it both new and interesting. Reviewer [8jNM] appreciates the comprehensive theoretical analysis and detailed explanation, as well as the clear and transparent contextualization relative to related work (although reviewer [UUBT] finds that related work is only sparingly discussed in main text)

**Weaknesses**: While the reviewers were overall appreciative of this submission, they also expressed concerns.

Reviewer [tFG1] notes that γ is always fixed in experiments, whereas it is typically optimized in VAEs. A comparison to a case with trainable γ would therefore be desirable. The reviewer also notes that it is unclear how big the gap is between ELBO and ELBO*.

Reviewer [UUBT] expresses concern that significance of the results is somewhat overstated. In particular it has been known that a linear-Gaussian VAE recovers the principal components and that the decoder therefore has orthogonal Jacobian columns (Dai et al 2018, Dai and Wipf 2019), and that nonlinear Guassian VAEs are also regularized towards decoders with orthogonal Jacobian columns. While no results for precise setting of work have been presented, this may reflect restrictiveness of setting, which assumed that data distribution has full support, along with an isotropic encoder (note: the authors clarified in their response that the encoder is in fact not isotropic).

Reviewer [8jNM] finds there is insufficient discussion of practical significance. In this context it would be helpful to understand IMA violation would be acceptable to still recover true factors. The reviewer further finds that the paper is dense and the appendix is long. The paper might be clearer if it focused more on one part of the story.

**Author Reviewer Discussion**: The authors provided detailed responses to all reviewers, including additional analyis of the role of the γ hyperparameter in response to [tFG1], additional discussion of the work by (Dai and Wipf 2019) in response to [UUBT], and provided various clarifications to reviewer [8jNM]. Reviewer [UUBT] raised their score 5->6 and reviewer [tFG1] raised their scored 6->7.

**Reviewer AC Discussion**: Reviewers [8jNM] and [tFG1] affirmed that they are happy for this paper to appear.

**Overall Recommendation**: The AC is satisfied with the level of examination and discussion that has taken place and will follow the recommendation of the reviewers. This is a relatively clear accept.

**Award:**

No

---

### Decision · Program_Chairs · 2022-09-14

Accept